# Microtubule disassembly by caspases is an important rate-limiting step of cell extrusion

Alexis Villars [1,2], Alexis Matamoro-Vidal[1], Florence Levillayer[1] & Romain Levayer [1✉]

The expulsion of dying epithelial cells requires well-orchestrated remodelling steps to maintain tissue sealing. This process, named cell extrusion, has been mostly analysed through the study of actomyosin regulation. Yet, the mechanistic relationship between caspase activation and cell extrusion is still poorly understood. Using the *Drosophila* pupal notum, a single layer epithelium where extrusions are caspase-dependent, we showed that the initiation of cell extrusion and apical constriction are surprisingly not associated with the modulation of actomyosin concentration and dynamics. Instead, cell apical constriction is initiated by the disassembly of a medio-apical mesh of microtubules which is driven by effector caspases. Importantly, the depletion of microtubules is sufficient to bypass the requirement of caspases for cell extrusion, while microtubule stabilisation strongly impairs cell extrusion. This study shows that microtubules disassembly by caspases is a key rate-limiting step of extrusion, and outlines a more general function of microtubules in epithelial cell shape stabilisation.

---

[1] Department of Developmental and Stem Cell Biology, Institut Pasteur, Université de Paris Cité, CNRS UMR 3738, 25 rue du Dr. Roux, 75015 Paris, France. [2] Sorbonne Université, Collège Doctoral, F75005 Paris, France. ✉email: romain.levayer@pasteur.fr

How epithelia maintain their physical and chemical barrier functions despite their inherent dynamics due to cell proliferation and cell death is a central question of epithelial biology. Cell extrusion, a sequence of coordinated remodelling steps leading to cell expulsion, is an essential process to conciliate high rates of apoptosis and cell elimination while preserving tissue sealing[1,2]. This process is essential for tissue homoeostasis and its perturbation can lead to chronic inflammation or contribute to tumoural cell dissemination[2,3]. Yet much remains unknown about extrusion regulation and orchestration.

Studies in the last decade have demonstrated that the remodelling steps of extrusions are mainly dependent on actomyosin contraction and mechanical coupling through E-cadherin (E-cad) adhesion. First, an actomyosin ring forms in the extruding cell driving cell-autonomous constriction[4–6]. This ring pulls on neighbouring cells through E-cad anchorage, resulting in force transmission which promotes the recruitment of actomyosin in the neighbouring cells and the formation of a supracellular actomyosin cable[1,4–7]. Eventually, the constriction of the cable combined with E-cad disassembly[6,8] leads to cell expulsion either on the apical or the basal side of the tissue. Meanwhile, neighbouring basal protrusions also contribute to cell detachment[9,10]. Alternatively, pulses of contractile medio-apical actomyosin can also contribute to cell expulsion[11,12]. Interestingly, while a lot of emphases has been given to actomyosin and E-cad regulation, we only have a limited understanding of the contribution of other cellular factors and cytoskeleton components to cell extrusion (see ref. [13] for one exception). Apoptosis is one of the main mode of programmed cell death which is essential for tissue homoeostasis and morphogenesis[14]. It is driven by the activation of caspases which through the cleavage of thousands of proteins orchestrate cell deconstruction[15]. While caspase activation is an important mode of epithelial cell elimination[14], how caspases orchestrate the key steps of extrusion remain poorly understood. Accordingly, only a handful of caspase targets relevant for cell extrusion have been identified so far[16].

The morphogenesis of the *Drosophila* pupal notum, a single-layer epithelium located in the back of the thorax, is an ideal system to study the regulation of apoptosis and cell extrusion. High rates of cell extrusion in reproducible patterns are observed in the midline and posterior region of the notum[17–21]. Interestingly, the majority of cell extrusion events in the pupal notum are effector caspase-dependent. Accordingly, caspase activation always precedes cell extrusion and inhibition of caspase in clones or throughout the tissue dramatically reduces the rate of cell extrusion[18,20–22]. However, we currently do not know which steps of extrusion are regulated by caspases or how effector caspase activation initiates and orchestrates epithelial cell extrusion.

Here, we performed the quantitative phenomenology of cell extrusion in the midline of the pupal notum. Surprisingly, while we observed the formation of a supracellular actomyosin ring in the late phase of extrusion, the initiation of cell apical constriction was not associated with any change in the dynamics and concentration of actomyosin and Rho. Accordingly, comparison with the behaviour of extruding cells in a vertex model suggested that cell extrusion in the notum is not initiated by a change of line tension. Moreover, the speed of extrusion is poorly affected by the reduction of MyoII activation. Instead, we found that cell extrusion initiation is concomitant with the disassembly of an apical mesh of microtubules (MTs). This disassembly is effector caspase-dependent and is required for cell extrusion. More importantly, the requirement of caspase activation for extrusion can be bypassed by MT disassembly, and MTs stabilisation strongly impairs cell extrusion, suggesting that the remodelling of MTs by caspases is an important rate-limiting step of cell extrusion. This work also emphasises the need to study the contribution of microtubules to epithelial cell shape regulation independently of actomyosin regulation.

## Results

**Actomyosin modulation is not responsible for extrusion initiation**. We focused on the *Drosophila* pupal notum midline (Fig. 1a, b), a region showing high rates of cell death and cell extrusion[17–20]. To better characterise the process of cell extrusion, we first quantified the evolution over time of the main regulators of cell–cell adhesion (E-cad) and cortical tension (non-muscle MyosinII, MyoII) by averaging and temporally aligning several extruding cells (one extrusion event lasting ~30 min). Contrary to other tissues[6,8], we did not observe a depletion of E-cad at the junctions during the constriction process, but rather a progressive increase of its concentration (Fig. 1c, d and Supplementary Movie 1). More strikingly, the onset of cell extrusion (defined by the inflexion of the apical perimeter, see "Methods") was not associated with a clear increase of MyoII levels (looking at myosin regulatory light chain, MRLC, Fig. 1e, f, Supplementary Movie 2), either at the junctional pool or in the medio-apical region (Supplementary Fig. 1a). Instead, a clear accumulation of MyoII forming a supracellular cable was observed during the last 10 min of cell extrusion (Fig. 1e, f and Supplementary Fig. 1a). Similar to extrusion in other systems[1,4,6,7], both the dying cells and its neighbours contribute to the late accumulation of MyoII, the cell-autonomous accumulation slightly preceding the accumulation in the neighbours (Supplementary Fig. 1b). Similarly, F-actin starts to accumulate only in the late phase of extrusion concomitantly with the formation of the supracellular cable (Fig. 1g, h, Supplementary Fig. 1h, i and Supplementary Movie 3), similar to the dynamics we observed for Rho1, a central regulator of F-actin, MyoII activity and pulsatility[23] (Supplementary Fig. 1j–l and Supplementary Movie 4).

Pulsatile actomyosin recruitment is observed during a wide variety of morphogenetic processes[24–26]. We also observed fluctuating levels of MyoII (Fig. 1i and Supplementary Fig. 1c) with pulses correlating with transient constriction of the cell apical perimeter (Supplementary Fig. 1c, d). The amplitude, duration and/or frequency of MyoII pulses can affect the efficiency of cell constriction[27]. However, we did not observe any significant change of these parameters before and after the onset of cell extrusion (Supplementary Fig. 1e–g). Finally, to better characterise the link between perimeter constriction and MyoII dynamics, we calculated a contraction yield (the ratio of constriction rate over the intensity of MyoII). We observed a significant increase in the contraction yield at the onset of cell extrusion (Fig. 1j, k), suggesting that similar MyoII pulses lead to more deformation after the onset of extrusion.

Altogether, we found that the initiation of cell extrusion and apical constriction is not associated with a significant change of actin, MyoII and Rho dynamics/levels. Their enrichment appears during the last 10 min of extrusion and is associated with the formation of a supracellular actomyosin cable. This suggests that MyoII activation/recruitment and dynamics are not sufficient to explain the initiation of extrusion and that MyoII activation is unlikely to be the rate-limiting step that initiates cell extrusion. Accordingly, we observed a significant increase of MyoII levels upon inhibition of caspase activity (by depleting Hid, a proapoptotic gene, Supplementary Fig. 1m, n), a condition that almost completely abolishes cell extrusion[20], suggesting once again that MyoII recruitment is not the main rate-limiting step of extrusion downstream of caspases.

**Cell extrusion in the midline is not driven by increased line tension**. To get a better understanding of the mechanical

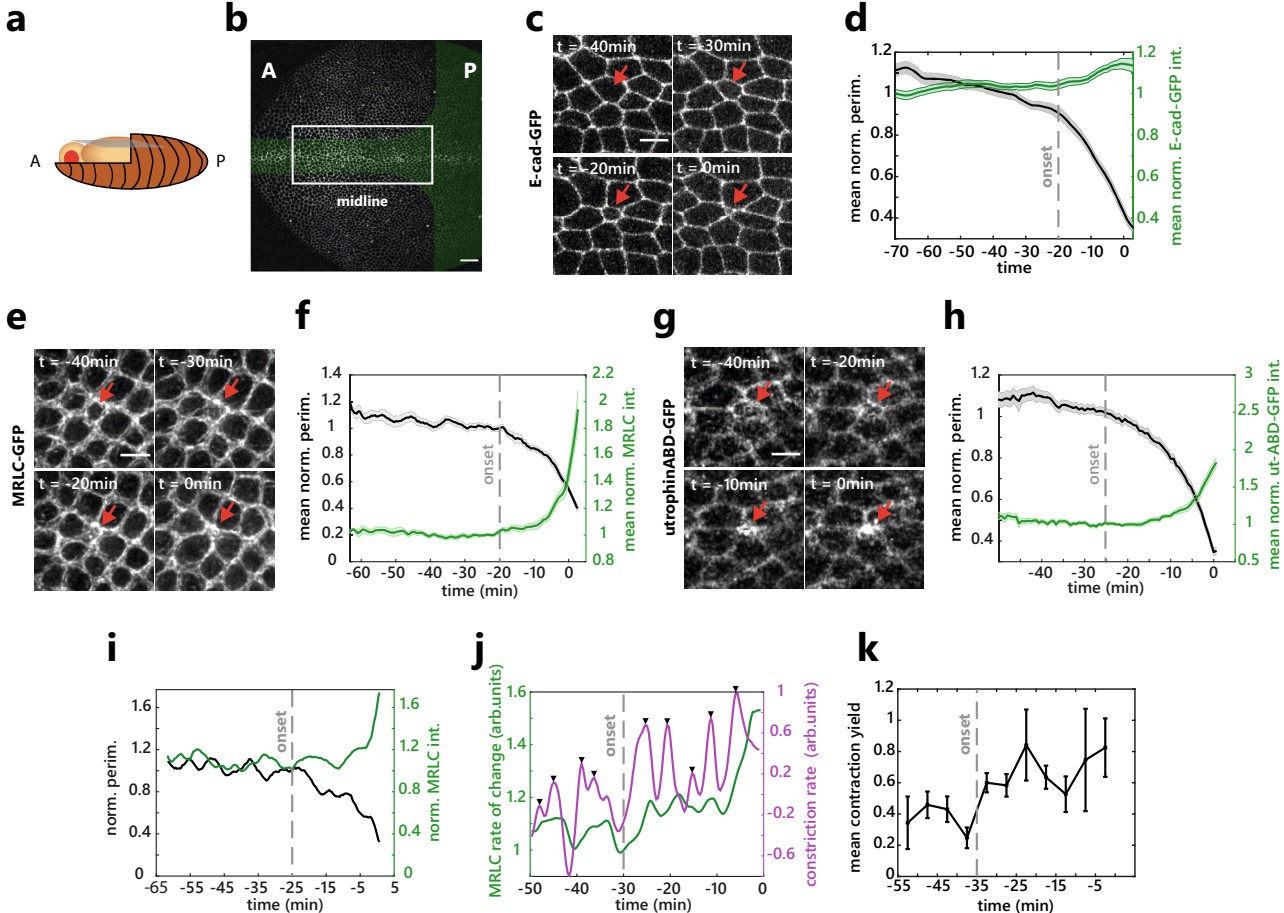

**Fig. 1 Actomyosin modulation is not responsible for the initiation of cell extrusion. a** Schematic of a *Drosophila* pupae. A: anterior, P: posterior. **b** Notum at 16 h after pupal formation (APF). Green zone: domains with a high rate of cell elimination, white rectangle: midline region. A: Anterior, P: Posterior. Scale bar, 25 μm. **c** Snapshots of E-cad-GFP during cell extrusion. The red arrow shows an extruding cell. $t_O$, time of extrusion termination (apical area = 0). Scale bar, 5 μm. **d** Averaged and normalised E-cad-GFP junctional signal during cell extrusion (green) and cell perimeter (black), light colour areas are SEM. The curves were aligned temporally by using the extrusion termination time point (see "Methods"). The grey dotted line represents the onset of extrusion marked by the inflection of the perimeter curve (see "Methods"). $N = 2$ pupae, $n = 27$ cells. **e** Snapshots of sqh-GFP (MRLC) during cell extrusion. The red arrow shows an extruding cell. $t_O$, time of extrusion termination. Scale bar, 5 μm. **f** Averaged normalised sqh-GFP (MRLC) total signal (medial+junctional) during cell extrusion (green) and cell perimeter (black). The grey dotted line is the onset of extrusion, light colour areas are SEM. $N = 2$ pupae, $n = 15$ cells. **g** Snapshots of actin during cell extrusion (utrophin Actin-Binding domain fused to GFP, utABD-GFP). Red arrow points at an extruding cell. $t_O$, time of extrusion termination. Scale bar, 5 μm. **h** Averaged normalised utABD-GFP total signal (medial + junctional) during cell extrusion (green) and cell perimeter (black). Grey dotted line represents the onset of extrusion, light colour areas are SEM. $N = 2$ pupae, $n = 37$ cells. **i** Single-cell representative curve of sqh-GFP (MRLC) total signal (green) and perimeter (black) showing MRLC pulsatility and perimeter fluctuations before and during cell extrusion. Grey dotted line represents the onset of extrusion. **j** Single-cell representative curve of sqh-GFP (MRLC) intensity rate of change (i.e., derivative, green) and the perimeter constriction rate (derivative of the perimeter, magenta). Black arrows show contraction pulses. Grey dotted line represents the onset of extrusion. **k** Averaged contraction yield (ratio of the constriction rate over MRCL junctional intensity) calculated in 5 min time windows (see "Methods"). The dotted line, extrusion onset, and error bars are SEM. $N = 2$ pupae, $n = 15$ cells. Source data are provided in the source data file.

parameters regulating the initiation of cell extrusion in the midline, we used a 2D vertex model. The apical area of cells in the model can be modulated by two main parameters: the line tension, a by-product of junctional actomyosin and cell–cell adhesion which tends to minimise the perimeter of the cell, and the area elasticity, which constrains the variation in the apical area of the cells and is thought to emerge from the incompressibility of cell volume and the properties of the medio-apical cortex[28,29]. The formation of a supracellular actomyosin cable, as observed in other instances of extrusion[1,6], should lead to an increase of line tension/contractility. Note that the line tension parameter in the vertex model do not distinguish the contribution of the dying cell and its neighbours. We simulated such extrusion by implementing a progressive increase of the contractility parameter in a single cell (see "Methods"). This led to a progressive decrease in

cell apical area concomitant with cell rounding (Fig. 2a, c, e, g and Supplementary Movie 5, progressive increase of cell circularity), in good agreement with the profile of extrusion observed in the larval epidermal cells of the abdomen which are driven by an actomyosin purse string[6] (Fig. 2i and Supplementary Fig. 2a). This, however, did not fit the pattern we observed in the midline of the pupal notum, where the initial phase of cell constriction is not associated with cell rounding (Fig. 2j). Cell constriction could also be initiated by a reduction of the cell resting area, which corresponds to the targeted apical area of cells in the absence of external constrains. Accordingly, the reduction of the resting area of a single cell in the vertex model led to cell constriction concomitant with a progressive reduction of cell roundness (Fig. 2a, d, f, h and Supplementary Movie 5, in good agreement with the dynamics we observed during the first 20 min of extrusion in the

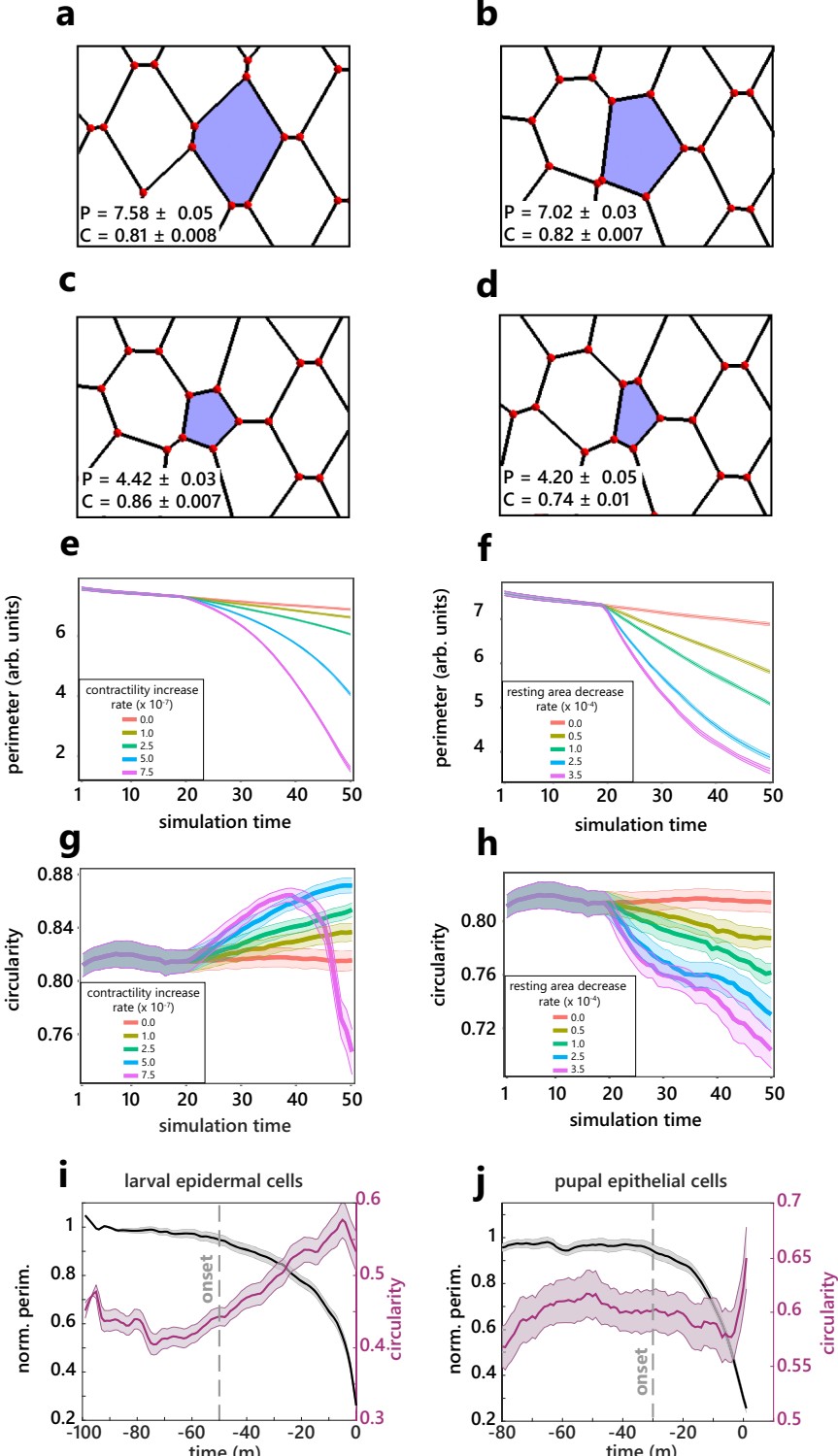

**Fig. 2 Cell extrusion in the midline is different from purse-string driven extrusion. a–d** Examples of a tracked cell (blue) from a vertex model during the early extrusion phase at different simulation time steps (sts) and under different conditions. $P$ is the average perimeter ± SEM, and C denotes average circularity ± SEM for the ten tracked cells of the simulations. **a** Initial state at sts = 1. **b** Control simulation (no change of parameter in the blue cell) at sts = 40. **c** Extrusion driven by increased cell contractility $\widetilde{\Gamma}_\alpha$ (contractility increase rate $c = 7.5.10^{-7}$), at sts = 40. **d** Extrusion through the decreased resting area $A\alpha^{(0)}$ (resting area decrease rate $r = 3.5.10^{-4}$) at sts = 40. **e, f** Averaged cell perimeter ± SEM for ten tracked cells as a function of contractility increase rate (**e**) and of resting area decrease rate (**f**). Variation of contractility and resting area was initiated at sts = 20. **g, h** Average cell circularity ±SEM for ten tracked cells as a function of contractility increase rate (**g**) and of resting area decrease rate (**h**). Variation of contractility and resting area was initiated at sts = 20. **i, j** Averaged cell circularity (magenta) and averaged and normalised cell apical perimeter (black) during cell extrusion in larval epidermal cells, n = 37 cells, N = 2 pupae. **i** and in the pupal notum epithelial cells, n = 22 cells, N = 2 pupae (**j**). Light colour areas are SEM. Grey dotted lines show the onset of extrusion. Source data are provided in the source data file.

notum (Fig. 2j, compare to yellow to purple lines in Fig. 2h with different rate of reduction of resting area). Importantly, while depleting ROCK (the main activator of MyoII[30]) by RNAi had a stringent effect on the epithelial morphology and blocked cytokinesis, it had no significant impact on the speed of extrusion (Supplementary Fig. 2b, c). In this context, the deformations occurring during extrusion were associated with a strong decrease of circularity and the absence of cell rounding in the late phase (Supplementary Fig. 2c), confirming the link between the late rounding observed in WT extrusion and the increase of actomyosin (Fig. 2j). Thus, global modulation of MyoII activation has little impact on cell extrusion in the notum, in agreement with an extrusion regime poorly relying on changes in line tension. Altogether, this confirmed that the initiation of cell extrusion in the midline is unlikely to be driven by an increase in line tension/junction contractility, but rather by a process modulating cell resting area/compressibility.

**The disassembly of microtubules correlates with the onset extrusion.** We next sought to identify which alternative factors could initiate cell extrusion in the midline of the pupal notum. Caspase activity can lead to a reduction of cell volume[31], which could be responsible for the reduction of cell apical area. However, we did not observe a significant change in cell volume during the process of cell extrusion (Supplementary Fig. 2d, e and Supplementary Movie 6). Alternatively, a downregulation of extracellular matrix (ECM) binding on the cuticle side could facilitate apical area constriction. Yet we did not observe a modulation of integrin adhesion components at the onset of extrusion (Supplementary Fig. 2f and Supplementary Movie 7). We, therefore, checked the distribution of microtubules (MTs), which can also regulate epithelial cell shape[32,33]. MT filaments accumulate in the medio-apical region of the midline cells as well as along the apicobasal axis (Fig. 3a–c). We tracked the orientation of MT growth in the medio-apical plane using the plus-end binding protein EB1, and found no obvious radial orientation of MTs (Supplementary Fig. 3a, b check also the non-extruding cells in Supplementary Movie 8), in agreement with a non-centrosomal pool of MTs[32]. Strikingly, we observed a significant and reproducible depletion of apical MTs at the onset of cell extrusion (visualised with the MT-associated protein Jupiter-GFP, Fig. 3d–f, Supplementary Movie 9). This depletion is concomitant with the onset of apical constriction (Fig. 3e, f, peak of cross-correlation between MTs intensity and cell perimeter with no significant lag time). The same downregulation was observed with EB1-GFP, a marker of MT plus ends (Supplementary Fig. 3c–g, Supplementary Movie 10, Supplementary Movie 8 at high temporal resolution), or upon expression of a tagged human α-tubulin (Supplementary Fig. 3h and Supplementary Movie 11). While we observed some fluctuations of MTs intensity in the non-extruding cells, the amplitude of these variations were much milder and non-persistent compared to the depletion observed during extrusion (see Fig. 3e inset and Supplementary Fig. 3g). Interestingly, MT depletion in the extruding cell was followed later-on by an accumulation in the neighbouring cells of MTs close to the junctions shared with the dying cell (Fig. 3d, g–i, Supplementary Fig. 3c–e and Supplementary Movies 9–11). This accumulation matches the timing of the actomyosin ring formation (Fig. 1e–h) and is reminiscent of the MTs reorganisation previously described near MDCK extruding cells[34]. The loss of apico-medial MTs may be driven by the reorganisation of the non-centrosomal MTs regulators Patronin and Shot[33,35–38]. However, we could not observe a clear modulation of their localisation/levels at the onset of cell extrusion (Supplementary Fig. 3i, j). Alternatively, the apical disappearance of MTs may be

driven by a basal shift of centrosomes (as observed in the dorsal folds of the early *Drosophila* embryo[33]), but we did not observe any change in the apicobasal position of the centrosomes at the onset of extrusion (Supplementary Fig. 3k). Finally, MTs down-regulation may be driven by a global disassembly of MT filaments. Accordingly, the disappearance of MT filaments is not restricted to the most apical domain and seems to occur throughout the cell, as visualised with αtub-mCherry (Supplementary Fig. 4a–c), or in single cells expressing EB1-GFP (Supplementary Fig. 4d, e). We then checked whether this depletion was dominated by a change of polymerisation rate or depolymerisation rate of MTs by assessing putative lag time between the changes of EB1 comets (new growing MTs) and changes in the total pool of MTs (using sirTubulin). We could not detect any significant lag time between the reduction of EB1 and total pool of MTs during extrusion (Supplementary Fig. 3l–n), which could be compatible with a process affecting both the polymerisation and the depolymerisation rate (see below and "Discussion"). Altogether, we observed a global disassembly of the MT network which is perfectly concomitant with the onset of cell extrusion and apical area reduction.

**MT depletion is effector caspase-dependent.** We then checked what could be responsible for MT disappearance. MT buckling driven by cell deformation can trigger MT disassembly[39–41]. As such, MT depletion during extrusion may be a cause or a consequence of the initiation of cell apical area constriction. To check if area constriction is sufficient to trigger MTs disassembly, we released tissue prestress in the notum by laser cutting a large tissue square[20,42]. The transient constriction of the cell apical area in the square region correlated with a transient increase of apical MT concentration (Supplementary Fig. 5a–d and Supplementary Movie 12), which is then followed by cell apical area re-expansion and MT intensity diminution. This suggests that cell apical constriction is not sufficient to disassemble MTs and that an active process must drive MT disassembly at the onset of extrusion. The activation of effector caspases is necessary for extrusion and always precedes cell constriction in the pupal notum[18,20,22]. We, therefore, checked whether effector caspase activation was necessary and sufficient for MT depletion. In agreement with caspase activation systematically preceding extrusion and MT depletion being concomitant with the onset of extrusion (this study), effector caspase activation (using the live marker GC3Ai[43]) was systemically observed in cells depleting MTs and extruding (Supplementary Fig. 5e and Supplementary Movie 13). We then checked whether caspase activation was sufficient to deplete MTs. Previously, we developed an optogenetic tool (optoDronc[21]) which can activate *Drosophila* Caspase9 triggering apoptosis and cell extrusion upon blue light exposure (Fig. 4a). We observed a rapid depletion of MTs (visualised with injected sirTubulin) upon activation of optoDronc in clones which was concomitant with cell apical constriction (Fig. 4b–d and Supplementary Movie 14, top, no significant lag time between sirTub diminution and perimeter constriction). Importantly, we confirmed in this condition that MTs depletion was occurring throughout the cell both on the apical and the basal side (Supplementary Fig. 4f, g). Activating optoDronc while inhibiting effector Caspases through p35 dramatically slowed down the rate of cell extrusion (ref. [21] and Fig. 4e–g and Supplementary Movie 14, bottom). While we could still see a late accumulation of MyoII in these slow constricting cells (Supplementary Fig. 5f, 3 h post optoDronc activation), we could no longer observe MT depletion but rather a progressive increase (albeit variable) in the apical concentration of MTs (Fig. 4e–g and Supplementary Movie 14, bottom). This is in agreement with the appearance of

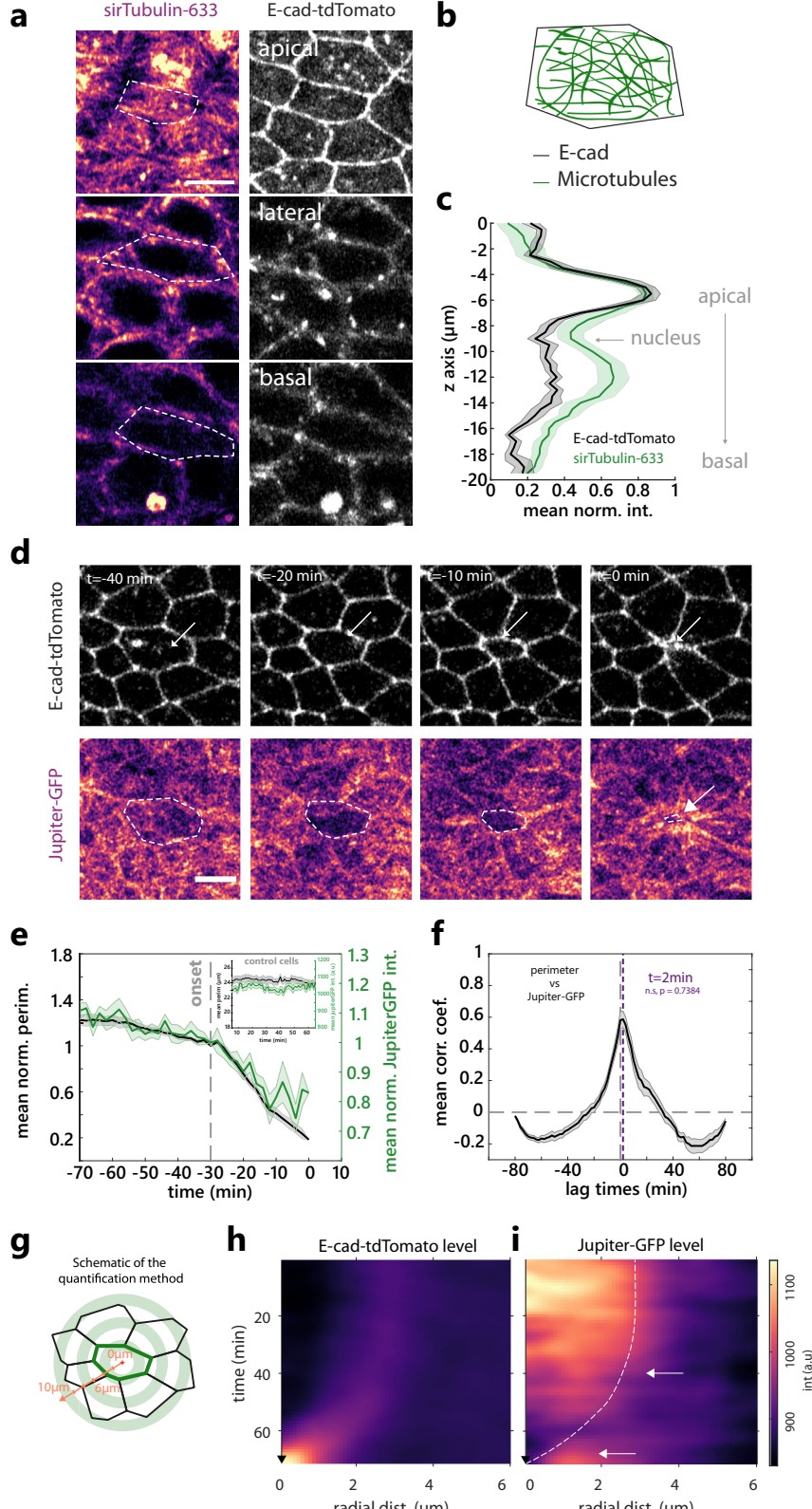

cells with the low apical area and strong tubulin accumulation that we observed in *hid-RNAi*/caspase-inhibited clones (Fig. 4h–j), a combination that we did not observe in WT cells. Interestingly, contrary to normal extrusions in the notum (Fig. 2j), circularity progressively increases during perimeter constriction upon optoDronc activation in cells expressing p35 (Fig. 4g). This fits with a slow constriction driven by an increase

of contractility/line tension. Altogether, this suggests that MT depletion during extrusion is driven directly or indirectly by effector caspases activation.

**MT up/downregulation can increase/decrease cell apical area**. The correlation between MT depletion and extrusion initiation

**Fig. 3 Microtubules depletion correlates with the onset of cell extrusion in the notum. a** Microtubules (MTs) visualised with sirTubulin (left, pseudocolour) and E-cad-tdTomato (right, greyscale) along the apicobasal axis in a midline cell. White dotted lines, cell contour. Scale bar, 5 μm. **b** Schematic of MTs orientation in the junctional plane. **c** sirTubulin (green) and E-cad-tdTomato (black) averaged and normalised intensity along the apicobasal axis. 0 μm is the most apical plane. $N = 10$ cells. The light area is SEM. **d** Snapshots of Jupiter-GFP (total tubulin, bottom, pseudocolour) and E-cad-tdTomato (top, greyscale) during cell extrusion, white arrows and white dotted lines show the extruding cell. $t_0$ min, termination of extrusion. Scale bar, 5 μm. **e** Averaged normalised Jupiter-GFP medial signal (green) and cell perimeter (black) during cell extrusion. Grey dotted line, extrusion onset, light colour areas, SEM. $t_0$ is the onset of extrusion. $N = 2$ pupae, $n = 24$ cells. Inset represents averaged perimeter (black) and Jupiter-GFP (green) intensity variations for control non-extruding cells. $N = 2$ pupae, $N = 21$ cells. **f** Averaged normalised cross-correlation of the cell perimeter vs Jupiter-GFP. The purple dotted line is at the maximum correlation coefficient ($t = 2$ min, not significantly different from $t = 0$ min, $P$ value $= 0.7384$, movie frame rate 1 min). Horizontal grey dotted line at correlation coefficient $= 0$. Vertical dotted line is at lag time $= 0$ min. Light area is SEM. $n = 24$ cells. **g** Schematic of the method used to represent averaged MTs intensity profile in space and time during extrusion using radial line intensity (orange line). Red cross shows the centre of the extruding cell (junctions, green). See "Methods". **h** Radial averaged kymograph (see panel **g**) of E-cad-tdTomato (left, pseudocolour), time is on the $y$ axis going downward, $x$ axis is radial distance from the cell centre. **i** Radial averaged kymograph of Jupiter-GFP signal. The white dotted line represents the average cell contour (maximum of E-cad average signal for each time point). Top white arrow points at the onset of Jupiter-GFP depletion. The bottom arrow shows Jupiter-GFP accumulation in the neighbouring cells at the end of the extrusion. $n = 24$ cells. Source data are provided in the source data file.

(Fig. 3), and the stabilisation of MTs observed upon caspase inhibition which prevents cell extrusion (Fig. 4), suggest that MT disassembly may be permissive for apical area constriction and extrusion initiation. Accordingly, previous studies in the *Drosophila* pupal wing and embryo have shown a role of medio-apical MTs in cell apical area stabilisation[32,33]. We, therefore, tested whether MTs could also modulate cell apical area in the pupal notum. We injected colcemid (a MT depolymerising drug) in pupae and assessed the efficiency of MT depletion through the disappearance of EB1-GFP comets and the inhibition of mitosis progression (Fig. 5a). UV exposure can locally inactivate colcemid[44]. Accordingly, local exposure to UV light (405 nm diode) led to a rapid recovery of EB1 comets (Fig. 5a–c, f and Supplementary Movie 15, bottom), although the initial organisation of MTs was not totally recovered. Strikingly, MT recovery was associated with a significant increase in cell apical area after few minutes, which was not observed in the control regions, or upon UV exposure in mock-injected pupae (Fig. 5e–h and Supplementary Movie 15, 4 min after the onset of UV exposure). This suggested that local MTs polymerisation is sufficient to increase cell apical area on a timescale of minutes. Importantly, the injection of colcemid did not lead to significant changes of MyoII levels during the first hours following injection, suggesting that these modulations are not driven by downstream effects on MyoII activation (Supplementary Fig. 6a–e).

We then tried to perform the reverse experiment (fast local depletion of MTs). Optogenetics can be used to trigger fast clustering and sequestration of proteins of interest. We used the LARIAT system (Light-Activated Reversible Inhibition by Assembled Trap) to trigger the clustering of a GFP-tagged α-tubulin upon blue light exposure[45–47]. Since the GFP-tagged α-tubulin knock-in is not viable[48], we instead used the over-expression of GFP-α-tubulin and LARIAT in clones to trigger a partial (endogenous tubulin is still present) depletion of α-tubulin. Accordingly, blue light exposure led to rapid clustering of GFP α-Tubulin, a mild reduction of sirTubulin signal (Supplementary Fig. 6f) and accordingly a mild but reproducible reduction of cell apical area without clear modulation of MyoII (Fig. 5i–k and Supplementary Movie 16). This apical area reduction was specific to tubulin sequestration since it was not observed upon clustering of a cytoplasmic GFP by LARIAT (Fig. 5k). Altogether, we concluded that a fast and local increase (or decrease) of MTs is sufficient to expand (or respectively constrict) cell apical area independently of noticeable MyoII modulation.

**The disassembly of MTs is an important rate-limiting step of extrusion**. We then checked whether MT depletion was sufficient

to trigger cell extrusion in the notum. Conditional induction of Spastin (a MT severing protein[49]) in clones was sufficient to deplete MTs (Supplementary Fig. 7a) and increase the rate of extrusion, including outside the midline in regions where no caspase activity is observed and where very few cells die in control conditions[18,20,21] (Fig. 6a–d and Supplementary Movie 17). To check whether MT depletion could indeed affect extrusion downstream of caspases, we assessed the impact of MT depletion on cells where caspase activation is inhibited. We used the inhibition of the proapoptotic gene *hid* which blocks caspase activation and cell extrusion in the notum[20]. While the depletion of Hid by RNAi drastically reduces the rate of extrusion[20], colcemid injection restored the rate of extrusion in *hid-RNAi* clones almost back to that of WT cells (Fig. 6e–g and Supplementary Movie 18). Importantly, while caspase activation almost systematically precedes cell extrusion in control conditions[18,20,21], a large proportion of cells underwent extrusion in the absence of caspase activation (visualised with GC3Ai[21,43]) in *hid RNAi* clones upon colcemid injection (Fig. 6h–j and Supplementary Movie 19). This suggested that MTs depletion can bypass the requirement of caspase activation for cell extrusion. Accordingly, while the inhibition of effector caspases (using UAS-p35) combined with Caspase9 activation (using optoDronc) drastically reduces the rate of extrusion and the cell constriction rate (Fig. 4f, g and Supplementary Movie 14, bottom), the rate of constriction was significantly enhanced upon MT depletion by colcemid injection (Fig. 6k, l and Supplementary Movie 20), albeit not back to WT speed. This confirmed that the accumulation of MTs we observed in optoDronc UAS-p35 clones (Fig. 4e, f) is one of the factors slowing down cell constriction and cell extrusion. To confirm that MTs depletion is indeed an important rate-limiting step of extrusion, we stabilised MTs using Taxol injection at a high concentration. While it did not totally block cell extrusion, it led to a global and drastic slowed-down of the speed of extrusion with variable durations (>threefolds increase of extrusion duration, Fig. 6m–o and Supplementary Movie 21). Moreover, the regime of deformation was now associated with a constant and significant increase of circularity (Fig. 6n), similar to the deformations observed upon increase of line tension (Fig. 2g), or upon optoDronc activation combined with p35 expression (Fig. 4g). Altogether, this demonstrates that the disassembly of MTs by effector caspases is an essential rate-limiting step of extrusion and one of the key initiators of cell constriction.

## Discussion

Our quantitative characterisation of cell extrusion in the pupal notum led to the surprising observation that actomyosin

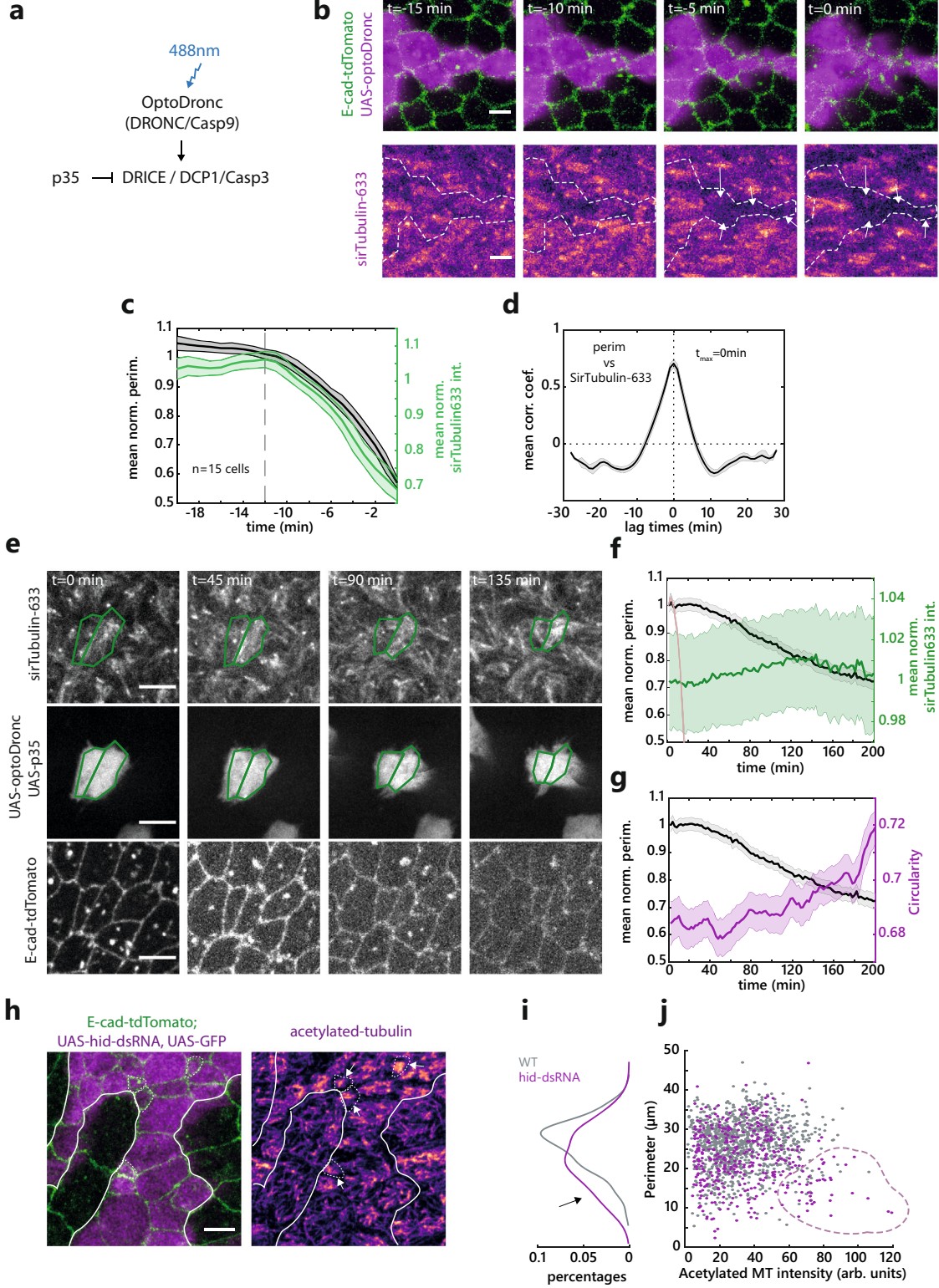

dynamics are not sufficient to explain the early steps of cell extrusion. We found instead that the disassembly of an apical MTs network correlates with the initiation of cell apical constriction and cell extrusion, which is then followed by more typical actomyosin ring formation. This is, to our knowledge, one of the first descriptions of a permissive role of MTs depletion in the initiation of cell extrusion independently of MyoII. MTs have been previously shown to influence cell extrusion through their reorganisation in the neighbouring cells and the restriction of

Rho activity toward the basal side[34]. Accordingly, we also observed an accumulation of MTs filaments in the neighbouring cells during the late phase of extrusion, which strikingly matches the timing of formation of a supracellular actomyosin ring. On the contrary, the novel function of MTs in the initiation of apical constriction that we described here is likely to be independent of Rho regulation. First, the initiation of cell deformation does not correlate with a change of actomyosin concentration/localisation/dynamics, nor recruitment of Rho. Second, cells inhibited

**Fig. 4 Microtubule depletion is effector caspase-dependent. a** Schematic of the caspase activation cascade, the effect optoDronc activation (caspase9) and p35 overexpression. **b** Snapshots of MTs depletion (SirTubulin, bottom, pseudocolour) upon activation of OptoDronc (Caspase9) by light. Top: E-cad-tdTomato (green) and a clone expressing OptoDronc-GFP (magenta). White dotted line, clone borders, white arrows, extruding cells. The scale bar is 5 μm. **c** Averaged and normalised medial apical sirTub signal (green) and perimeter (black) during cell extrusion induced by optoDronc. Light colour areas, SEM. Grey dotted line, onset of extrusion. $N = 1$ pupae, $n = 15$ cells. **d** Averaged normalised cross-correlation of the cell perimeter vs sirTub. The maximum correlation coefficient is a $t = 0$ min. The light area is SEM. $n = 15$ cells. **e** Snapshots of UAS-optoDronc-GFP, UAS-p35 cells slowly contracting upon blue light exposure. Top row, sirTub, middle row, cells expressing UAS-OptoDronc-GFP and UAS-p35, bottom row, E-cad-tdTomato. Scale bars are 5 μm. **f, g** Averaged and normalised medial apical sirTub signal (green), cell perimeter (black) (**f**) and averaged circularity (**g**, magenta) during the slow constriction of cell expressing UAS-OptoDronc-GFP and UAS-p35 upon blue light exposure. The pink curve shows the normal speed of extrusion upon optoDronc activation for comparison (see panel **c**). $N = 2$ pupae, $n > 99$ cells. Light colour areas are SEM. **h** z-projection of cells stained for acetylated tubulin (pseudocolour, right) and E-cad (green) inside and outside a clone depleted for the proapoptotic gene *hid* (UAS-hid-dsRNA, magenta). White lines: clone contour. White arrowheads, abnormal small cells in the clone accumulating acetylated tubulin (white dotted lines, cell contours). The scale bar is 5 μm. Representative of four experiments. **i, j** Distribution of perimeter values for WT (grey) and UAS-hid-dsRNA cells (purple), percentage of the total cell population (**i**). Cloud plot of acetylated-α-tubulin levels (x axis) as a function of cell perimeter (y axis), one dot for one cell (**j**). Black arrow (**i**) and purple dotted area (**j**) highlight the population of small cells with high levels of acetylated-α-tubulin present only in the UAS-hid-dsRNA population. $N = 5$ pupae, $N = 1002$ WT cells and $N = 405$ hid-dsRNA cells. Source data are provided in the source data file.

for caspases activation do not extrude, despite the significant accumulation of MyoII (upon depletion of Hid, Supplementary Fig. 1m, n or using optoDronc combined with p35, Fig. 4e–g, Supplementary Fig. 5f). Third, the evolution of apical cell shape during the early phase of extrusion does not match the evolution expected through an increase of line tension (unlike the extrusion of larval epidermal cells, Fig. 2). Fourth, the depletion of ROCK had a very minor effect on the speed of extrusion (Supplementary Fig. 2b, c), while MT stabilisation through Taxol injection led to a drastic slow-down of extrusion (Fig. 6m–o). Thus, the effect of MT stabilisation on extrusion cannot be explained by an indirect effect through the sequestration of Rho activators. Altogether, this strongly argues for an initiation of extrusion which is independent of the formation of an actomyosin purse string and a minor role of the modulation of line tension. As such, our study outlines a novel role of MTs in epithelial cell shape stabilisation which is independent of MyoII regulation. Recently, several works have described the central role of non-centrosomal MTs in epithelial morphogenesis, however, this was mostly through their impact on MyoII activity[35–37]. Our study reinforces the notion that MTs may also stabilise cell apical area independently of MyoII regulation, as previously shown during morphogenesis of the *Drosophila* embryo[33] or of the *Drosophila* pupal wing[32]. Interestingly, the accumulation of apical acetylated MTs promotes the capacity of cells to re-insert in an epithelial layer through radial intercalation in *Xenopus*[50,51]. This nicely mirrors the function of MTs disassembly that we found in cell extrusion.

Through which mechanisms could MTs stabilise cell apical area? MTs are well known for their stabilising function of cell membrane protrusions during cell migration[52] and can also modulate single-cell major-axis length[53]. Indeed, MTs embedded in the actomyosin network can bear significant compressive forces[54,55] and modulate cell compressibility[56], or bear the compression driven by the constriction of cardiomyocytes[57]. Thus, apical MTs could directly resist the pre-existing cortical tension in the midline cells and their disassembly would be sufficient to trigger cell constriction. Alternatively, MTs may influence the contractile properties of the actomyosin cortex independently of Rho activity and MyoII phosphorylation, hence modulating cell deformation without apparent changes in actomyosin recruitment. Accordingly, MTs disassembly is sufficient to accelerate the kinetics of actomyosin constriction in vitro[58]. Finally, MTs may have a more indirect function either by modulating nuclei positioning[59,60], hence releasing space to facilitate apical constriction, or by directly modulating cytoplasmic viscosity[61].

We found that MTs disassembly is driven directly or indirectly by effector caspases. The disappearance of apical MTs is not driven by a shift of the centrosome position[33] (Supplementary Fig. 3k), nor relocalisation of MTs or a change of MTs association with the centrosome[35], or modulation of the localisation of the non-centrosomal MTs organisers Patronin and Shot[33,36,37] (Supplementary Fig. 3i, j). The disassembly occurs instead throughout the cell (Supplementary Fig. 4) and may be driven by a global modulation of core MTs components by caspases. The disassembly of MTs could either be driven by a reduction of the polymerisation rate, an increase of the depolymerisation rate or both. In principle, this could be sorted by comparing the number of EB1 positive comets relative to the total number of MTs over time. However, we found that EB1-GFP signal and total MT signal (using sirTubulin) decrease concomitantly during extrusion with no significant lag time (Supplementary Fig. 3l–n). While we cannot exclude that our limited temporal and spatial resolution may miss subtle time differences, this is compatible with a mechanism affecting both the polymerisation and the depolymerisation rate, as would happen upon sequestration and destruction of tubulin monomers. Accordingly, α-tubulin and β-tubulin are both cleaved by caspases in S2 cells and these cleavages are conserved in humans[62]. This is in agreement with the depletion that we also observed with the tagged human α-tubulin (Supplementary Fig. 3h). However, we could not address the functional relevance of these cleavages since the mutant form of α-tubulin (mutation at the three cleavage sites) did not integrate properly in MT filaments either in S2 cells or in the notum (Supplementary Fig. 8). Since several core MT components are targets of caspases (including α-tub and β-tub)[62], we believe that the inhibition of the caspase-dependent disassembly of MTs will be hard to achieve. Moreover, we cannot exclude at this stage alternative mechanisms of MTs destabilisation based on the modulation of plus-end binding proteins or crosslinkers by caspases. Of note, the redundancy of multiple caspase targets triggering MT depletion and the high conservation of several cleavage sites may reflect the physiological importance of this regulatory process.

We showed previously that caspase activation is required for cell extrusion in the pupal notum, including during cell death events induced by tissue compaction[18,20,21]. This suggested that cell extrusion is unlikely to occur spontaneously upon cell deformation and that permissive regulatory steps are required to allow cell expulsion. Our work suggests that the disassembly of MTs by caspases may be one key rate-limiting step. Accordingly, MTs depletion is sufficient to bypass the requirement of caspase activity for cell extrusion (Fig. 6). Interestingly, several

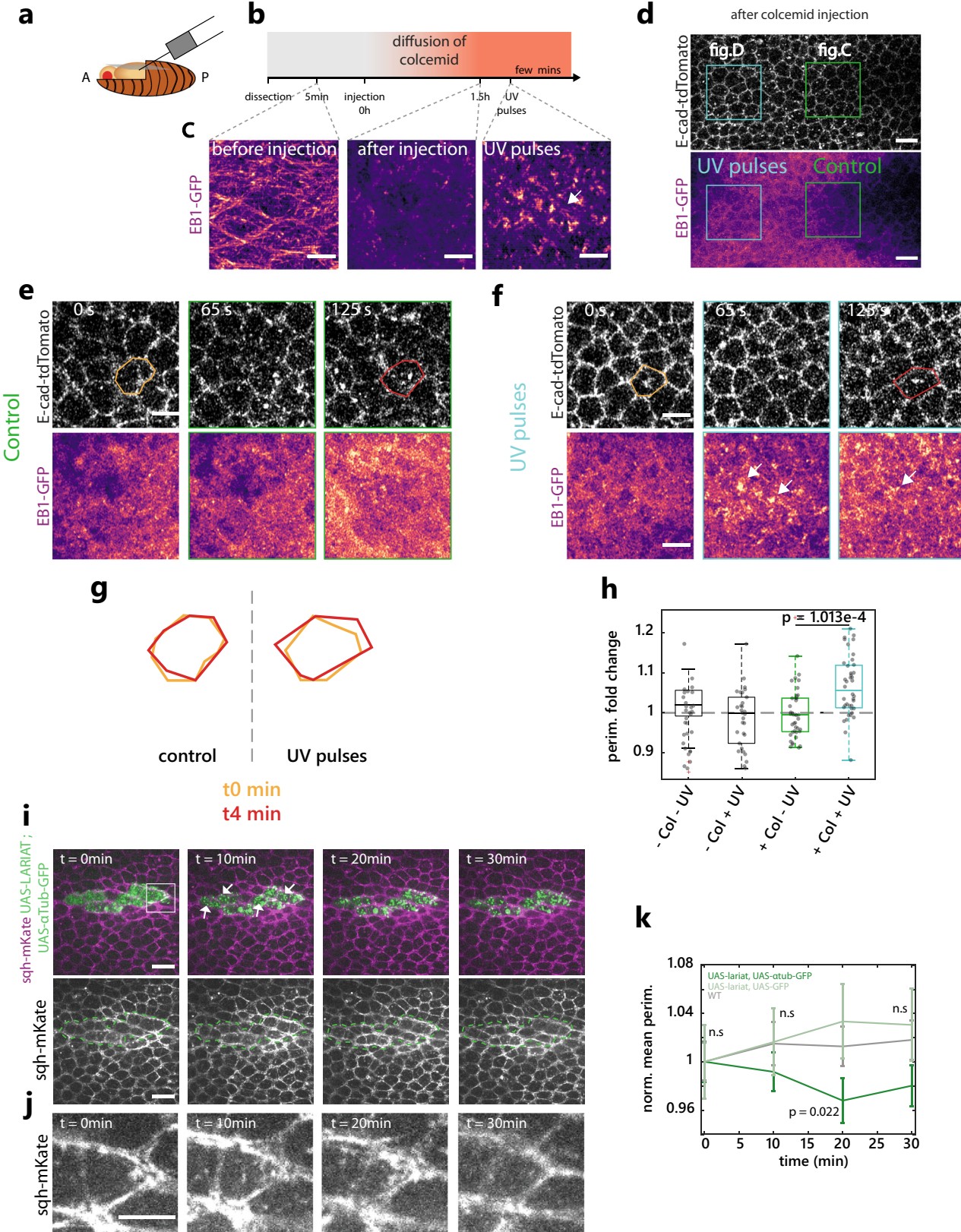

mechanisms of MTs repair and stabilisation upon mechanical stress have been recently characterised[39,40,63], including recent works characterising the mechanism of MT stabilisation upon compression[64,65], which can reinforce the capacity of MTs to bear mechanical load[57]. Thus, stress generated by cell constriction and/or tissue compression is unlikely to be sufficient to trigger

MT disassembly. This is in good agreement with the transient accumulation of MTs that we observed upon tissue stress release (Supplementary Fig. 5a–c and Supplementary Movie 12), and outlines the requirement for an active disassembly mechanism by caspases to trigger MT depletion and extrusion. The stabilisation of MT upon cell constriction may help to buffer variations of the

**Fig. 5 MTs polymerisation/depolymerisation can increase/decrease cell apical area. a** Schematic of the injection in the *Drosophila* pupae. **b, c** Timeline of the injection protocol and colcemid inactivation experiment (see "Methods") and EB1-GFP signal (pseudocolour) before injection, 1.5 h after injection and after few minutes of colcemid inactivation by UV. Scale bars are 5 μm. **d** Snapshot of the different experimental regions after colcemid injection, single plane. Top, E-cad-tdTomato, bottom, EB1-GFP (pseudocolour, note the disappearance of MT filament). Green, control region shown in (**e**). Blue, UV-exposed region shown in (**f**). Scale bar, 10 μm. **e, f** Snapshots of a control region (**e**) and UV-exposed region (**f**) after colcemid injection. Top, E-cad-tdTomato. Bottom shows EB1-GFP (pseudocolour). Scale bars are 5 μm. Orange and red contour, single representative cell at $t_0$ and 4 min. **g** Examples of cell area evolution, orange $t_0$, red 4 min post UV exposition in the control region (left) or UV-exposed region (right). **h** Quantification of the cell perimeter fold changes. Col: colcemid. Green boxplot, colcemid without UV pulses, blue boxplot, colcemid and UV. $N = 4$ colcemid injected pupae, 3 control pupae, $n > 30$ cells per condition (one dot = one cell). Each boxplot shows the median value (middle line), and 25th and 75th percentiles. The whiskers show the most extreme data points not considered outliers. Outliers are plotted in red. *P* value, two-sided *t* test. **i** sqh-mKate (MRLC, purple, greyscale bottom) and a clone expressing UAS-LARIAT and UAS-αTubulin-GFP (green dotted line, clone contours). Arrows show GFP clusters forming after 488-nm exposure. $t_0$, start of the movie. The white box highlights the cell shown in (**j**). The scale bar is 10 μm. **j** Snapshot of sqh-mKate in a cell upon LARIAT activation. Scale bar is 5 μm. **k** Normalised averaged perimeter upon LARIAT clusterisation of αTubulin-GFP (dark green) compared to control cells (grey, outside the clone) or in control cells expressing UAS-LARIAT and UAS-GFP cytoplasmic (light green). $N = 3$ pupae for each, $n = 41$ WT and UAS-Lariat, UAS-αTubulin-GFP cells $n = 30$ UAS-lariat, UAS-GFP cells per time points. Error bars are SEM. n.s. not significant. *P* value, paired *t* test. Source data are provided in the source data file.

cell apical area despite the fluctuations of line tension[26] or external mechanical constrains. In absence of MTs, this negative feedback would be gone, hence allowing large fluctuations of cell area and the appearance of spontaneous extrusion driven by local mechanical instabilities[66]. Interestingly, depletion of MTs in other systems (e.g. the fly embryo[35]) does not seem to have the same impact on epithelial stability and cell extrusion, suggesting that the impact of MTs on epithelial cell stabilisation may be context-dependent. Finally, it should be noted that MT disassembly in caspase-inhibited cells does not completely rescue the speed and the rate of extrusion, thus other unidentified targets of caspases are likely to also participate in extrusion regulation.

Caspase activation in the pupal notum, and in other tissues, does not lead systematically to cell extrusion and cell death[18,20,21,67]. The mechanisms downstream of effector caspase activation governing cell survival or engagement in apoptosis remain poorly understood. Since the engagement of cells in extrusion in the WT notum systematically leads to cell death[17,18], and since MT depletion is the earliest remodelling step associated with extrusion, the disassembly of MTs by caspases is likely to be one of the key decision steps, leading to engagement in apoptosis in the pupal notum. Future work connecting cell mechanical state, quantitative caspase dynamics and MT remodelling may lead to important insights about the decision of a cell to die or survive.

## Methods

### Experimental model and subject details

**Drosophila melanogaster *husbandry*.** All the experiments were performed with *Drosophila melanogaster* fly lines with regular husbandry technics. The fly food used contains agar agar (7.6 g/l), saccharose (53 g/l), dry yeast (48 g/l), maize flour (38.4 g/l), propionic acid (3.8 ml/l), Nipagin 10% (23.9 ml/l) all mixed in one litre of distilled water. Flies were raised at 25 °C in plastic vials with a 12 h/12 h dark-light cycle at 60% of moisture unless specified in the legends and in the table below (alternatively raised at 18 °C or 29 °C). Females and males were used without distinction for all the experiments. We did not determine the health/immune status of pupae, adults, embryos and larvae, they were not involved in previous procedures, and they were all drug and test naive.

**Drosophila melanogaster *strains*.** The strains used in this study and their origin are listed in Table 1.
The exact genotype used for each experiment is listed in Table 2.

### Generation of α-tub84B-mCherry WT and non-cleavable mutant. 

The inserts mCherry-alphaTub84b and mCherry-alphaTubD34A-D48A-D200A (mutation of the three caspase cleavage sites) were generated by PCR from pAc-mCh-Tub (Addgene 24288) with oligos containing mutations or not. The triple mutant at the three sites (mCherry-alphaTubD34A-D48A-D200A) was generated by using the following primers combination (see table below for primer sequence): F1 + R1, F2 + R2, F3 + R3. The WT form was generated using the F1 + R3 primers. The PCR products were then inserted in the pJFRC4-3XUAS-IVS-mCD8::GFP

(Addgene 28243) linearised by NotI, XbaI digestion, using NEBuilder HiFi DNA Assembly Method. The construct was checked by sequencing and inserted at the attp site attp40A after injection by Bestgene. The primers used for the construct are listed below (inserted mutation sites are shown in red).

| Primers | sequence |
|---------|----------|
| F1 | taaccctaattcttatcctttacttcaggcggccgcaacatggtgagcaagggcgagga |
| F2 | agatgccgtctgacaagaccgtgggcggaggtgatgCctcgttcaacaccttcttcagc |
| F3 | ccctggagcattccgCctgcgccttcatggtcgaca |
| R1 | ctccgcccacggtcttgtcagacggcatctggccaGcgggctggatgccgtgctc |
| R2 | accatgaaggcgcagGcggaatgctccagggtggtg |
| R3 | acagaagtaaggttccttcacaaagatcctctagattagtactcctcagcgccct |

**S2 cell culture.** S2R + (DGRC stock 150, RRID:CVCL_Z831) cells were cultured in Schneider's *Drosophila* Medium with 10% foetal bovine serum, penicillin and streptomycin. S2R + cell were transfected with FUGENE HD (Promega, ref: E2311). Twenty-four hours after transfection, S2R + cells were plated on glass-bottom dishes coated with concanavalin A (con A). Cells were imaged in a spinning disk confocal 30–60 min after cell spreading on the dishes.

**Immunostaining.** Dissection and immunostainings of nota were performed as indicated in ref. [68] with standard formaldehyde fixation and permeabilisation/washes in PBT 0.4% Triton. The following antibody was used: mouse anti-acetylated-Tubulin (1/200, Sigma T7451, clone 611B1). A secondary antibody was produced in goat with Alexa 633 (1/100, Life technologies, A21052, lot 1712097). Dissected nota were mounted in Vectashield with DAPI (Vectorlab).

**Vertex modelling of cell extrusion.** To model the early steps of extrusion, we used a computational vertex model based on the existing computational framework for the study of developmental processes in the epithelial tissues of *Drosophila*[29,69]. The model was implemented in gfortran, using openGL to visualise the outputs.

In the vertex model, only the apical sides of the cells are considered. Cells are represented as 2D polygons, made of vertices connected by edges. The vertices can move over time as a result of intra- and intercellular mechanical forces. The movement of the vertices is implemented by comparing the mechanical energy of a vertex in its current position $(x, y)$ with the energy of a randomly chosen point nearby $(x+\delta d, y+\delta d)$ with $\delta d \in [0,0.005]$. When the energy in the new position is smaller, then the movement is accepted as the new vertex location. When the energy is bigger, the movement is accepted with probability $P_{accept}$ (=0.05) in order to introduce stochastic fluctuations.

The energy (E) of a vertex i is given by

$$E(R_i) = \sum_\alpha \frac{K_\alpha}{2}(A_\alpha - A_\alpha^{(0)})^2 + \sum_{(i,j)} \Lambda_{ij} \cdot l_{ij} + \sum_\alpha \frac{\Gamma_\alpha}{2} \cdot L_\alpha^2 \qquad (1)$$

where $R_i = (x_i, y_i)$ is the position of the vertex i. The first and the third summations are over all the cells α in which the vertex i is present, and the second summation is over all the cell edges $[i, j]$ in which the vertex i is present. $A_\alpha$ is the apical area of the cell α and K is the area elasticity modulus, which is assumed to be equal for all the cells in our simulations. $A_\alpha^{(0)}$ is the resting area of the cell α. The distance and the line tension between the pairs of vertices $[i, j]$ are denoted $l_{ij}$ and $\Lambda_{ij}$ respectively. The third term includes the perimeter of the cell α ($L_\alpha$) and the perimeter contractility coefficient ($\Gamma_\alpha$). By choosing $\sqrt{A\alpha^{(0)}}$ as a unit of length and $(KA\alpha^{(0)})^2$ as a unit of energy (as in ref. [69]), dividing both sides of Eq. (1) by

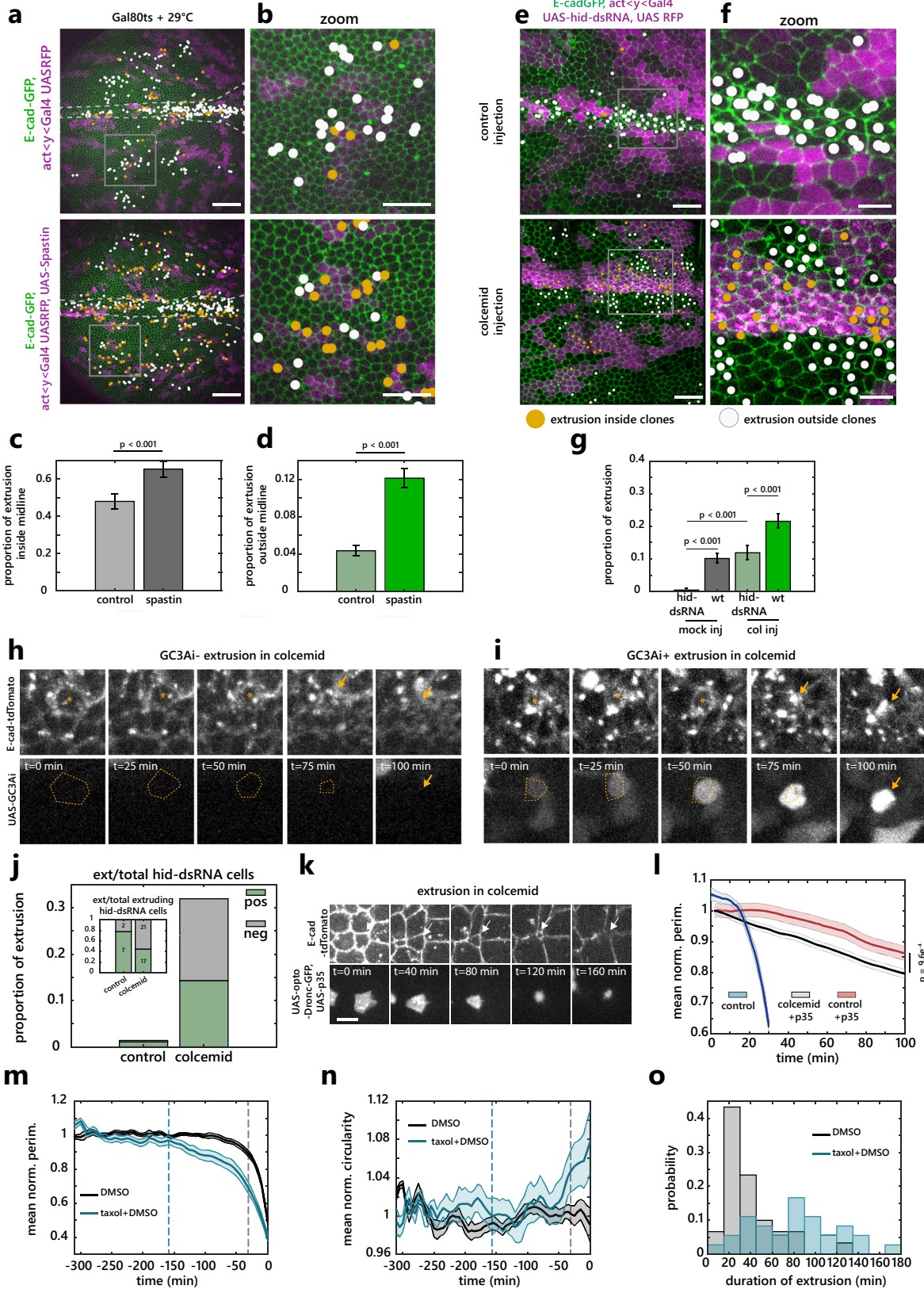

$(KA\alpha^{(0)})^2$ results in the following dimensionless equation:

$$\widetilde{E}(R_i) = \sum_{\alpha} \frac{1}{2}\left(\frac{A_\alpha}{A\alpha^{(0)}} - 1\right)^2 + \sum_{(i,j)} \widetilde{\Lambda}_{ij} \cdot \frac{l_{ij}}{\sqrt{A\alpha^{(0)}}} + \sum_{\alpha} \frac{\widetilde{\Gamma}_\alpha}{2} \cdot \frac{L_\alpha^2}{A\alpha^{(0)}} \qquad (2)$$

Where $(A_\alpha/A\alpha^{(0)})$, $(l_{ij}/\sqrt{A\alpha^{(0)}})$ and $(L_\alpha^2/A\alpha^{(0)})$ are, respectively, dimensionless area, bond length and perimeter. This model is characterised by dimensionless line

tension $(\widetilde{\Lambda}_{ij} = \Lambda_{ij}/K(A\alpha^{(0)})^{\frac{3}{2}})$ and dimensionless perimeter contractility $(\widetilde{\Gamma}_\alpha = \Gamma/KA\alpha^{(0)})$ that were set respectively to 0.06 and 0.02 as in ref. [70].

Rearrangements of the topology of the vertices (T1 transitions) were allowed when two vertices $i$, $j$ were located less than a minimum distance $d_{min}$ (= 0.2) apart, and a movement of one of the vertices was energetically favourable such that the distance between the vertices decreases.

**Fig. 6 The disassembly of MTs by caspases is an important rate-limiting step of extrusion. a, b** E-cad-GFP (green) in nota with control clones (magenta, top) or clones expressing conditionally Spastin (magenta, bottom). Coloured dots, extrusions inside (orange) or outside clones (white). White dotted lines, midline. White rectangles, regions are shown in (**b**). Scale bars, 50 μm (**a**) and 25 μm (**b**). **c, d** Proportion of cell elimination throughout the nota (see "Methods") over 1000 min, inside (**c**) or outside (**d**) the midline. Error bars, 95% confidence interval, P values, Fisher exact test. Control, $N = 2$ pupae, 630 midline and 5105 outside-midline cells, Spastin, $N = 3$ pupae, 993 midline and 6997 outside-midline cells. **e, f** Nota expressing E-cad-GFP (green) and hid-dsRNA clones (magenta) injected with EtOH+$H_2O$ (top, control) or colcemid (bottom). Scale bars, 25 μm (**e**), 10 μm (**f**). White rectangles, regions are shown in (**f**). **g** Proportion of cell elimination inside hid-dsRNA clones and outside (wt) in EtOH+$H_2O$ (left) or colcemid injected (right) pupae over 500 min throughout the notum. P values, Fisher exact test. Error bars, 95% confidence interval. $N = 4$ control pupae, 155 WT and 1150 hid-dsRNA cells, $N = 3$ colcemid pupae, 1388 WT and 926 hid-dsRNA cells. **h, i** hid-dsRNA cell extrusions after colcemid injection negative (**h**) or positive (**i**) for effector caspase activity (GC3Ai, bottom). Yellow dotted lines, extruding cell contour. Scale bars, 5 μm. **j** Proportion of hid-dsRNA cell extrusion positive or negative for GC3Ai in mock ($n = 9$ extrusions, 2 pupae) or colcemid injected pupae ($n = 39$ extrusions, 2 pupae). Inset, relative proportions and absolute numbers. **k** optoDronc, p35 extruding cell (white arrows) upon colcemid injection (E-cad-tdTomato, top, optoDronc-GFP bottom). Scale bar = 5 μm. **l** Normalised averaged perimeter of cells expressing optoDronc and p35 upon blue light exposure in controls (red, Fig. 4e–g) or upon colcemid injection (black). Blue curve, optoDronc only, from Fig. 4c. Light colour areas, SEM. $n > 99$ cells, $N = 3$ control and $N = 2$ colcemid pupae. **m, n** Normalised averaged perimeter (**m**) and circularity (**n**) of extruding cells upon taxol injection (blue, taxol in DMSO) or DMSO alone (black). Dotted lines, onset of extrusion. Light colour areas, SEM. **o** Distribution of extrusion duration in Taxol (blue) or DMSO alone injected pupae (black). $N = 2$ pupae, $n = 30$ control extruding cells, $n = 36$ taxol extruding cells. Source data are provided in the source data file.

**Table 1 Drosophila melanogaster strains used in this study.**

| Fly line | Chromosome location | Origin (citation) | RRID |
|---|---|---|---|
| E-cad-tdTomato (KI) | II | [76] | BDSC_58789 |
| E-cad-GFP (KI) | II | [76] | BDSC_60584 |
| hs-flp22; sqh-sqh-GFP; sqh-sqh-GFP | II, III | Thomas Lecuit | Thomas Lecuit |
| Sqh-utABD-GFP | II | [24] | Thomas Lecuit |
| ubi-aniRBD-GFP | II | [23] | Thomas Lecuit |
| Pnr-gal4 | III | Bloomington | BDSC_3039 |
| sqh-Sqh-3XmKate2 | II | [77] | Yohanns Bellaïche |
| hs-flp22; sqh-Sqh-3XmKate2; act < y < gal4, UAS-BFP | I, II, III | Yohanns Bellaïche | Yohanns Bellaïche |
| Hs-flp22; act < y < gal4, UAS-mcd8RFP/Cyo | I, II | Our group | Our group |
| Hs-flp22;; act < y < gal4, UAS-nlsRFP/TM6b | I, III | Bloomington | BDSC_30558 |
| Ubi-E-cad-GFP | II | DGRC | DGRC_109007 |
| UAS-hid ds-RNA (III) | III | VDRC GD 8269 | VDRC |
| UAS-p35 | III | Bloomington | BDSC_5073 |
| UAS-EB1-GFP | III | Bloomington | BDSC_35512 |
| Jupiter-GFP | Trap, III | Antoine Guichet | BDSC_6836 |
| Ubi-Talin-GFP | X | Guy Tanentzapf[78] | Guy Tanentzapf |
| UAS-Spastin | III | Antoine Guichet | Antoine Guichet |
| Ubi-Patronin-GFP | II | Bloomington | BDSC_55128 |
| shot(L)-GFP (TRAP) | II | DGRC | DGRC_109707 |
| Ubi-Sas4-GFP | II | Renata Basto | FBal0240464 (flybase ID) |
| UAS-human-αtub-mCherry | II | Bloomington | BDSC_25774 |
| UAS-αtub-GFP | III | Bloomington | BDSC_7253 |
| UAS-vhhGFP-CRY2-CIBN | II | [79] | Xiaobo Wang |
| UAS-optoDronc-GFP | II (attp40) | [21] | Our group |
| sqh-Sqh-3XmKate2 | II | [77] | Yohanns Bellaïche |
| tub-Gal80ts | II | Bloomington | BDCS_7019 |
| hs-flp22; Ubi-Ecad-GFP, UAS-RFP; act < y < gal4 | II | Loïc le Goff | Loïc le Goff |
| UAS-GC3Ai | III | [43] | Magalie Suzanne |
| UAS-his3.3 mIFP-T2A-H01 | III | Bloomington | BDSC_64184 |
| Hs-flp22;; Act < cd2 < gal4,UAS-GFP | I; III | [20] | [20] |
| UAS-Cd4-mIFP-T2A-H01 | III | Bloomington | BDSC_64182 |
| UAS-α-tubulin-mCherry | II | This study | This study |
| UAS-α-tubulinmutABC-mCherry | II | This study | This study |
| endocadGFP, tub-Gal80ts/Cyo; Act < y < Gal4 UAS-nls-RFP/TM6B | II, III | Our group | Our group |
| UAS-Rok- dsRNA | II | VDRC | VDRC 3793GD |
| Sqh > sqhGFP, FRT40A/Cyo GFP | II | [80] | Yohanns Bellaiche |
| FRT40A, sqhRFP[247]/Cyo | II | [80] | Yohanns Bellaiche |
| E-cad::3xmKate (KI)/Cyo GFP | II | [77] | Yohanns Bellaiche |

To avoid buckling at the boundary of the tissue, we assumed a greater stiffness of the cells edges located at the external boundary of the tissue, and set that the line tension for the external edges was higher ($= 1.6.\tilde{\Lambda}_{ij}$) than that of the internal edges.

Simulations started from a tissue of 1141 cells, among which 10 cells scattered in the tissue (far from the edge to avoid boundary effects) were tracked for circularity ($C = 4.\pi$ (area/perimeter$^2$)) and perimeter (P) changes during the simulation run. In the initial topology, tracked cells have an average P of 7.58 ± 0.05 (SEM) and an average C of 0.81 ± 0.088 (Fig. 2a). Simulations were run for 6.5 million iterations, one iteration consisting in moving a randomly chosen vertex, updating its energy, and deciding to accept the movement or not. For clarity, the simulation run was divided in 50 simulation time steps (sts), with 1 sts = 130.000 vertex iterations.

**Table 2 Precise genotype used for each figure panel.**

| Figure | Genotype | Clone induction | Dev. time |
|---|---|---|---|
| 1b | E-cad-GFP(KI); pnr-Gal4 | – | 16 h APF |
| 1c, d | hs-flp22; ubi-E-cad-GFP UAS-RFP/+; act<y < Gal4/UAS-hid-dsRNA (quantification outside clones) | – | 16 h APF |
| 1e, f, i–k; S1 a, c–g | hs-flp22; sqh-Sqh-GFP; sqh-Sqh-GFP | – | 20 h APF |
| S1b | hsFLP; sqh-Sqh-GFP, FRT40A/ FRT40A, sqh-Sqh-RFP | 48 h ACI, 12 min hs | 16 h APF |
| 1g, h, S1h, i | hs-flp; act<y < gal4, UAS-mcd8RFP/Cyo; sqh-GFP-utABD/TM6b | – | 16 h APF |
| S1j–l | w; E-cad-tdTomato (KI)/+; +/ubi-mE-AnilRBDGFP | – | 16 h APF |
| S1m, n | hs-flp22; sqh-sqh::GFP/Cyo; act<cd2 < gal4, UAS-nlsRFP/UAS-hid-dsRNA | 48 h ACI, 12 min hs | 20 h APF |
| 2i | w; E-cad-GFP(KI) | – | |
| 2j | w; E-cad-tdTomato(KI)/+;pnr-Gal4/UAS-EB1-GFP | – | |
| S2a top | hs-flp22; sqh-Sqh::GFP; sqh-Sqh::GFP | – | 20 h APF |
| S2a bottom | w; E-cad-GFP(KI) | – | 20 h AFP |
| S2b, c | w; E-cad::3xmKate(KI); pnr-Gal4/UAS-ROK-dsRNA | – | 16 h APF |
| S2d, e | w; UAS-optoDroncCRY2-GFP/ E-cad-tdTomato; MKRS/act<y < Gal4 UAS-GFP | 48 h ACI, 12 min hs | 16 h APF |
| S2f | ubi-Talin::GFP/FM7 | – | 16 h APF |
| 3a, c | w; E-cad-tdTomato(KI) | – | 16 h APF |
| 3d–i | w; E-cad-tdTomato(KI)/cyo; Jupiter-GFP/TM6B | – | 16 h APF |
| S3a–g & l–n | w; E-cad-tdTomato(KI)/+; pnr-Gal4/UAS-EB1-GFP | – | 16 h APF |
| S3h | w; E-cad-GFP(KI)/+; pnr-Gal4/UAS-alpha-Tub-mCherry-human | – | 16 h APF |
| S3i | w; E-cad::3xmKate (KI)/shot(l)-GFP (TRAP) | – | 16 h APF |
| S3j | w;ubi-p63-Patronin.A.GFP/Cyo | – | 16 h APF |
| S3k | w; E-cad-tdTomato(KI)/ubi-Sas4-GFP | – | 16 h APF |
| S4a–c | w; E-cad-GFP(KI)/UAS-α-tubulin-mCherry; pnr-Gal4/+ | – | 16 h APF |
| S4d–e | hs-flp22; sqh-sqhmKatex3/+; act<y < Gal4, UAS-BFP/UAS-EB1-GFP | 48 h ACI, 12 min hs | 16 h APF |
| S4f, g | w; UAS-optoDroncCRY2-GFP/ E-cad-tdTomato(KI); MKRS/act<y < Gal4 UAS-GFP | 48 h ACI, 12 min hs | 16 h APF |
| 4a–d | w; UAS-optoDroncCRY2-GFP/ E-cad-tdTomato(KI); MKRS/act<y < Gal4 UAS-GFP | 48 h ACI, 12 min hs | 16 h APF |
| 4e–g | w; UAS-optoDroncCRY2-GFP/ E-cad-tdTomato(KI); UAS-p35/act<y < Gal4 UAS-GFP | 48 h ACI, 12 min hs | 16 h APF |
| 4h–j | hs-flp22; E-cad-tdTomato(KI)/Cyo; act<y < Gal4, UAS-GFP/UAS-hid-dsRNA | 48 h ACI, 12 min hs | 20 h APF |
| S5a–d | w; E-cad-GFP(KI)/UAS-α-tubulin-mCherry; pnr-Gal4/+ | – | 30 h APF |
| S5e | w; UAS-α-tubulin-mCherry/UAS-GC3AI; pnr-Gal4/+ | | 30 h APF |
| S5f | w; sqh-sqh-mKatex3/ UAS-optoDroncCRY2-GFP; act<y < Gal4, UAS-BFP/UAS-p35 | 48 h ACI, 12 min hs | 30 h APF |
| 5a–h | w; E-cad-tdTomato(KI)/+;pnr-Gal4/UAS-EB1-GFP | – | 16 h APF |
| 5i–k | hs-flp22; sqh-sqh-mKatex3/UAS-vhhGFP-CRY2-CIBN; act<y < Gal4, UAS-BFP/UAS-αTubulin-GFP | 48 h ACI, 12 min hs | 16 h APF |
| 5k | hs-flp22; E-cad-mKatex3(KI)/UAS-vhhGFP-CRY2-CIBN; act<y < Gal4, UAS-GFP/+ | 48 h ACI, 12 min hs | 16 h APF |
| S6 a–e | w; sqh-sqh-mKatex3/Cyo; pnr-gal4/TM6b | – | 16 h APF |
| S6f | hs-flp22; sqh-sqh-mKatex3/UAS-vhhGFP-CRY2-CIBN; act<y < Gal4, UAS-BFP/UAS-αTubulin-GFP | 48 h ACI, 12 min hs | 16 h APF |
| 6a–d | hs-flp22; tub-Gal80ts/ ubi-E-cad-GFP, UAS-RFP; UAS-Spastin/act<y < Gal4 | 96 h ACI 18 °C, 12 min hs | 16 h APF |
| 6a–d | hs-flp22; tub-Gal80ts/ ubi-E-cad-GFP, UAS-RFP; MKRS/act<y < Gal4 | 96 h ACI 18 °C, 12 min hs | 16 h APF |
| 6e–g | hs-flp22; ubi-E-cad-GFP, UAS-RFP/+; act<y < Gal4/UAS-hid-dsRNA | 48 h ACI, 12 min hs | 16 h APF |
| 6h–j | hs-flp22; E-cad-tdTomato(KI)/UAS-GC3AI; act<y < Gal4, UAS-His3-mIFP/UAS-hid-dsRNA | 48 h ACI, 12 min hs | 16 h APF |
| 6k, l | hs-flp22; E-cad-tdTomato(KI)/ UAS-optoDroncCRY2-GFP; act< y < Gal4, UAS-GFP/UAS-p35 | 48 h AIC, 12 min hs | 16 h APF |
| 6m–o | w; E-cad::3xmKate(KI)/+; pnr-Gal4/UAS-EB1-GFP | – | 16 h APF |
| S7a | hs-flp22; endocadGFP, tub-Gal80ts/+; Act<y < Gal4 UAS-nls-RFP/UAS-Spastin | 96 h ACI 18 °C, 12 min hs | 16 h APF |
| S8c, d | w; E-cad-GFP(KI)/UAS-α-tubulin-mCherry-ABC; pnr-Gal4/+ | – | 16 h APF |
| S8c, d | w; E-cad-GFP(KI)/UAS-α-tubulin-mCherry; pnr-Gal4/+ | – | 16 h APF |

*ACI* time after clone induction, *APF* after pupal formation, *hs* heat shock at 37 °C.

At $t = 20$ sts, three different conditions were examined to test for the effect of the mode of extrusion on cells circularity during the early stages of extrusion. In the first condition, the ten tracked cells had parameter values identical to all the other cells of the tissue (control). In the second condition, the ten tracked cells were forced to initiate extrusion by increasing at each iteration their contractility parameter ($\tilde{\Gamma}$) with a fixed rate $c$ ($\tilde{\Gamma}_{t+1} = \tilde{\Gamma}_t + c \cdot \tilde{\Gamma}_t$) thus simulating a purse-string driven extrusion. Five different values were examined for $c$, with $c = [0.0; 1.0; 2.5; 5.0; 7.5] \times 10^{-7}$. In the third condition, the ten tracked cells were forced to initiate extrusion by decreasing after each iteration their resting area ($A\alpha^{(0)}$) with a fixed rate $r$ ($A\alpha^{(0)}_{t+1} = A\alpha^{(0)}_t - r \cdot A\alpha^{(0)}_t$). Simulations were run for five $r$ values ($r = [0.0; 0.5; 1.0; 2.5; 3.5] \times 10^{-4}$).

**Optogenetic control**

*Induction of cell death using optoDronc.* For induction of optoDronc in clones in the pupal notum, *hs-flp; E-cad-tdTomato(KI); act < cd2 < G4, UAS-GFP* females

were crossed with homozygous *UAS-optoDronc* or *UAS-optoDronc; UAS-p35*. Clones were induced through a 12 min heat shock in a 37 °C water bath. Tubes were then maintained in the dark at 25 °C. White pupae were collected 48–72 h after clone induction and aged for 16 h at 25 °C in the dark. Collection of pupae and dissection were performed on a binocular with LED covered with a homemade red filter (Lee colour filter set, primary red) after checking that blue light was effectively cut (using a spectrometer). Pupae were then imaged on a spinning disc confocal (Gataca system). The full tissue was exposed to blue light using the diode 488 of the spinning disc system (12% AOTF, 200 ms exposure per plane, 1 stack/min). Extrusion profiles were obtained by segmenting extruding cells in the optoDronc clones with E-cad-tdTomato signal in the notum using the Fiji plugin Tissue analyzer[71] (https://github.com/baigouy/tissue_analyzer). Curves were aligned on the termination of extrusion (no more apical area visible) and normalised with the averaged area on the first five points. The same procedure was used upon control injection or injection of colcemid (see below) in optoDronc UAS-p35 background, except that curves were aligned to $t_0$, the onset of blue light

exposure. Note that in this condition, all cells of the clones were segmented irrespective of the size of the clone, which can affect by itself the speed of extrusion[21].

*LARIAT mediated depletion of αTubulin-GFP.* UAS-vhhGFP-CRY2-CIBN (hereafter LARIAT) was expressed in gal4-expressing clones. CRY2 dimerises with CIBN in a light-dependent manner. The association with the anti-GFP nanobody (vhhGFP) allows to trap the αtubulin-GFP in these large clusters. Clones were induced through a 12 min heat shock in a 37 °C water bath. Tubes were then maintained in the dark at 25 °C. White pupae were collected 48–72 h after clone induction and aged for 16 h at 25 °C in the dark. Pupae were then dissected an imaged using the same method than described for the optoDronc condition. Quantification were made by segmenting manually cells from clones and in control population at four time points ($t = 0, 10, 15, 20$ min). The same protocol was applied in the LARIAT control expressing cytoplasmic GFP.

**Live imaging and movie preparation.** Notum live imaging was performed as followed: the pupae were collected at the white stage (0 h after pupal formation), aged at 25° or 29°, glued on double-sided tape on a slide and surrounded by two home-made steel spacers (thickness: 0.64 mm, width 20 × 20 mm). The pupal case was opened up to the abdomen using forceps and mounted with a 20 × 40 mm #1.5 coverslip where we buttered halocarbon oil 10 S. The coverslip was then attached to spacers and the slide with two pieces of tape. Pupae were collected 48 or 72 h after clone induction and dissected usually at 16 to 18 h APF (after pupal formation). The time of imaging for each experiment is provided in the table above. Pupae were dissected and imaged on a confocal spinning disc microscope (Gataca systems, Metamorph software) with a ×40 oil objective (Nikon plan fluor, N.A. 1.30) or ×100 oil objective (Nikon plan fluor A N.A. 1.30) or a LSM880 (Zen black software) equipped with a fast Airyscan using an oil ×40 objective (N.A. 1.3) or ×63 objective (N.A. 1.4), Z-stacks (0.5 or 1 μm/slice), every 5 min or 1 min using autofocus at 25 °C. The autofocus was performed using E-cad signal as a plane of reference (using a Zen Macro developed by Jan Ellenberg laboratory, MyPic) or a custom-made Metamorph journal on the spinning disc. Movies were performed in the nota close to the scutellum region containing the midline and the aDC and pDC macrochaetae. Movies shown are adaptive local Z projections. Briefly, E-cad plane was used as a reference to locate the plane of interest on sub-windows (using the maximum in Z of average intensity or the maximum of the standard deviation) through the Fiji plugin LocalZprojector or corresponding MATLAB routine[72].

**Laser ablation.** Photo-ablation experiments were performed using a pulsed UV-laser (355 nm, Teem photonics, 20 kHz, peak power 0.7 kW) coupled to a Ilas-pulse module (Gataca systems) attached to our spinning disk microscope. The module was first calibrated and then set between 30 and 40% laser power to avoid cavitation. Images were taken every 1 min and ablation started after one time point. In all experiments, 400 × 400 μm rectangle were converted to line of 10 thickness in metamorph. Repetitions were set between 5 and 10 for proper cut to be achieved. Cell perimeter was obtained through cell segmentation and the tubulin signal quantified in the total area of each cell (contour +3px).

**Image processing and inflection point detection.** All images were processed using Matlab R2020a and FIJI (http://fiji.sc/wiki/index.php/Fiji). Movies for analysis were obtained after local Z projections of z-stacks using the Fiji LocalZprojector plugin[72]. As we were interested in apical signals, we set $\Delta Z = 1$ so 3 planes of 0.5 μm or 1 μm were projected using maximum intensity projections. Then extruding cells were manually detected. When needed the signal was corrected for slight bleaching using CorrBleach macro from EMBL (https://www.embl.de/eamnet/html/bleach_correction.html). In order to measure signal intensities, single cells were segmented using E-cad signal when it was possible (otherwise sqh or utABD signal). Depending on the data this was done directly using Tissue analyzer Fiji plugin after local z projections or after using EPySeg (https://github.com/baigouy/EPySeg)[73]. Once segmented, ROI of the cell contour was extracted to Fiji and custom macro was used to measure the mean px intensities of medio-apical signal (−3px from the junction), total signal (+3px from the contour) or junctional signal (transformation to an area of 6px wide encompassing the junctions). The perimeter was measured using the real cell contour. In addition, the noise was removed for spinning disc movies when needed by subtracting the signal measured in a 20 × 20-px ROI outside the tissue. Results were then analysed in MATLAB.

For analysis, single curves were aligned either by the end (i.e., the moment of the end of extrusion) or by the inflection point of the perimeter. Inflection points were automatically detected using a homemade MATLAB function. Briefly, it uses two moving linear fits after smoothing the perimeter using MATLAB moving average (taking into account five data point windows). The point with minimal error between the two fits and real data corresponds to the point where the perimeter starts to constrict i.e., the inflection point. Average, standard deviation and SEM were calculated after the alignment.

Radial averaged kymograph were obtained by tracking the centroid of every extruding cell and measuring the intensity along concentric circles of 3px at different distances from the cell centre. The values were averaged for every tracked extruding cell. E-cad signal was used to detect cell contour and define MTs signal from the extruding cell and from the neighbours.

*Quantifications along the apicobasal axis.* The cell average intensity per plane was quantified in extruding cells −45, −30 and −15 min before the end of extrusion using a 20 × 20-px square over the whole cell with a resolution of $z = 0.5$ μm. Z axis profiles were aligned by the maximum value of E-cad-GFP to correct for small variation in z in the z-stack and align to the apical plane. For each cell, intensities were normalised by a min-max normalisation to make comparison possible after the removal of background noise. P values are obtained by pairwise and single-tailed t tests.

For the single-cell example, the apical area was measured by segmenting the cell. The intensity of apical MT was measured in two consecutive apical planes. The basal intensity was measured in the two consecutive planes below the nucleus. The represented side views are selected where we observe the largest cross-section of the nucleus in this cell.

**Myosin peak detection and yield computation and cross-correlation.** First, the signal was smoothed using a rolling average of 0.07 (using the LOESS option in the smooth method in MATLAB) in order to filter for noise. We then computed the contraction rate as following $-\frac{d\,perimeter}{dt}$ and the myosin rate of change as $\frac{d\,myosin}{dt}$. In order to assess how closely changes in myosin relate to constriction, we computed the cross-correlation between myosin level or myosin rate of change and the constriction rate. The cross-correlation was calculated on Matlab with the *xcorr* function with the "*coef*" option (normalised cross-correlation after subtracting the mean). All the curves (one per cell) were then aggregated and averaged.

Contraction peak and myosin peak were detected using the findpeaks function in Matlab by setting "MinPeakProminence" to 7 in order to filter for noise. The yield was calculated for each contraction peak of each single curve as follow $\frac{contraction_{T\,peak}}{myosin\,int_{Tpeak}}$. Perimeter and intensity curves were temporally aligned and normalised by their inflection point and yield data were sorted relative to the onset in 5-min time windows. Then single curves were aggregated and averaged. In order to compute myosin pulse duration, amplitude and frequency we detected myosin pulses. We then returned the peak parameters: $T_{peak}$ (Time at peak maximum), $W_{peak}$ (width at half peak maximum) and $A_{peak}$ (peak amplitude). Myosin pulse frequency was computed as followed: $\frac{N_{peak}}{Total\,Time_{before\,onset}}$ or $\frac{N_{peak}}{Total\,Time_{after\,onset}}$. Finally, all curves were aligned based on lag times (lag time = 0 min) for the averaging.

**Injection in pupal notum.** In all, 16 h APF pupae were glued on double-sided tape and the notum was dissected with a red filter. Pupae were then injected using homemade needles (borosilicate glass capillary, outer diameter 1 mm, internal diameter 0.5 mm, 10 cm long, Sutter instruments) pulled with a Sutter instrument P1000 pipette pulling apparatus. Injections were performed in the thorax of the pupae using a Narishige IM400 injector using a constant pressure differential (continuous leakage). Colcemid (Sigma Demecolcine D7385 5 mg) was diluted in ethanol to a stock concentration of 20 mM and was then injected at 2.5 mM in the thorax of the pupae. Note that some cells seem to enlarge (see for instance Fig. 6k) in the neighbouring cells) due to entry in mitosis and failure to complete mitosis. Taxol (Sigma Taxol/Paclitaxel T7402, 5 mg) was diluted in DMSO to a stock concentration of 60 mM (50 mg/mL) and was then injected at this concentration in the thorax of the pupae. SirTubulin was diluted in DMSO to a stock solution of 1 mM and then diluted to a concentration of 100 μM and injected in the thorax.

**Colcemid inactivation by UV and effect of MT re-polymerisation on cell area.** Pupae were dissected and mounted as described above. We first imaged EB1-GFP to assess its dynamics prior to colcemid injection. Then pupae were unmounted and we injected colcemid as described in this protocol. We then waited for 1 hour and 30 min for colcemid to diffuse in the notum. Next, we re-imaged EB1-GFP and assessed colcemid effect through the loss of EB1-GFP comets and diffusion of the GFP pool as well as cell division arrest (cell division arrest was later used to assess colcemid effect whenever we could not image EB1-GFP signal).

We then image a single z-plane every second for 240 s. We inactivated colcemid locally by pulsing 405 nm diode (0.44% AOTF) in a restricted region of the imaging field and compared this to a control region. We assessed MT re-polymerisation by looking at the formation of EB1-GFP foci and comets. In order to measure the effect of re-polymerisation on cell size, we segmented the cell in the UV or control regions at $t_0$ and $t_{240s}$. As we are interested in the relative changes of cell perimeter between these two time points we computed a perimeter fold change for each cell as the following ratio $\frac{perim_{t240}}{perim_{t0}}$ and compared the condition with colcemid to a control condition injected with $H_2O$ + ethanol.

**Proportion of cell elimination in clones**

*Spastin clones.* Since we could not recover clone upon Spastin overexpression, we generated clones allowing conditional induction of Spastin with act<y < Gal4 UAS-RFP, UAS-Spastin with tub-Gal80[ts]. Gal80[ts] binds Gal4 and represses Gal4-driven expression. Upon switching to 29 °C Gal80[ts] becomes inactive allowing Gal4-driven expression.

Due to the maturation time of RFP following the temperature shift to 29 °C, it is difficult to track the position of the clones initially following the temperature shift. For that reason, we decided to measure the proportion of cell extrusion by looking at the global rate of cell extrusion at 29 °C in the condition with control UAS-RFP

of UAS-Spastin, UAS-RFP clones. Thus, we tend to underestimate the real increase in extrusion rate. Because of the high rates of cell extrusion in the midline we separated the quantification between the inside of the midline and outside. We did that by tracing manually the midline using the position of the most central Sensory Organ Precursors which define the midline. We then manually detected all the extrusions over 1000 min and defined automatically if they belong to the midline or not. We then segmented the tissue at $t_0$ to count the number of cells inside the midline or outside and then used these values to compute the proportion of extrusion as following: $\frac{N_{extrusion\ midline}}{N_{cell\ midline}}$ or $\frac{N_{extrusion\ outside}}{N_{cell\ outside}}$.

*Rescue of extrusion in hid-dsRNA following colcemid injection.* For these experiments, colcemid was injected as described above and this condition was compared with control injections ($H_2O$ + ethanol). We then manually detected all the extrusions at each time points and for each condition during the 500 first minutes. Clones were segmented at each time point using the UAS-RFP signal and we used this segmentation to automatically define if extrusions belong to UAS-hid-dsRNA clones or not. We then segmented the tissue at t0 min to count the number of cells in the clones or outside the clones and used these values to compute the proportion of extrusion inside or outside the clone as following $\frac{N_{extrusion\ clones}}{N_{cell\ clones}}$ or $\frac{N_{extrusion\ wt}}{N_{cell\ wt}}$.

*Detection of Caspase signal in hid-dsRNA clones.* Colcemid was injected as described previously in this protocol and this condition was compared with control injections ($H_2O$ + ethanol). We then manually detected all the extrusions at each time point and for each condition and manually defined if they are positive or negative for caspase activation using the UAS-GC3AI-GFP signal. We then either computed the proportion of each "type" of extrusion relative to the total number of cell in the clones or to the total number of extrusions in the clones.

**Microtubule stabilisation and delay of extrusion**. Taxol was injected as explained previously (see methods injection) and results were compared to pupae injected with DMSO only. To make the comparison possible, all cells were aligned by the end and normalised by the average perimeter value before the onset of extrusion (or similarly by the average circularity value before the onset).

The duration of extrusion was measured as the difference between the time at the end of extrusion and the time at the inflexion point. The represented distribution is normalised by the total number of observed extrusions in each condition.

**Statistics**. Data were not analysed blindly. No specific method was used to pre-determine the number of samples. The definition of n and the number of samples is given in each figure legend and in the table of the Experimental model section. Error bars are standard error of the mean (SEM). P values are calculated through $t$ test if the data passed the normality test (Shapiro–Wilk test), or Mann–Whitney test/rank-sum test if the distribution was not normal, or the Fisher exact test for comparison of proportion (see legends). Statistical tests were performed on Matlab.

**Reporting summary**. Further information on research design is available in the Nature Research Reporting Summary linked to this article.

## Data availability

All the data generated in this study (images, local projections of movies) have been deposited on Zenodo with the following link https://doi.org/10.5281/zenodo.654683[74]. The quantified data used for every graph are provided in a single excel source data file (one sheet per figure panel). Further information and requests for resources and reagents should be directed to and will be fulfilled by the corresponding author, Romain Levayer (romain.levayer@pasteur.fr). All the reagents generated in this study will be shared upon request to the corresponding author without any restrictions. Source data are provided with this paper.

## Code availability

All codes generated in this study have been deposited on Github and can be accessible with the following link https://doi.org/10.5281/zenodo.654530[75]. Further explanations and help can be provided upon request to the lead contact.

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

## Acknowledgements

We thank members of RL lab for critical reading of the manuscript. We would like to thank Jakub Voznica for his observations on the abdomen during his internship. We are also grateful to Antoine Guichet, Thomas Lecuit, Magalie Suzanne, Yohanns Bellaïche, Xiaobo Wang, Renata Basto, the Bloomington Drosophila Stock Centre, the Drosophila Genetic Resource Centre and the Vienna Drosophila Resource Centre, Flybase for sharing essential information, stocks and reagents. We also thank Benoît Aigouy for the Tissue Analyser software and Jan Ellenberg group for MyPic autofocus macro. A.V. is supported by a PhD grant from the doctoral school "Complexité du Vivant" Sorbonne Université and from an extension grant of La Ligue contre le Cancer, work in RL lab is supported by the Institut Pasteur (G5 starting package), the ERC starting grant CoSpaDD (Competition for Space in Development and Disease, grant number 758457), the Cercle FSER and the CNRS (UMR 3738).

## Author contributions

R.L. and A.V. discussed and designed the project and wrote the manuscript. A.M.V. performed the vertex simulations. F.L. designed the tubulin mutant construct. A.V. performed all the other experiments and analyses. Every author has commented and edited the manuscript.

## Competing interests

The authors declare no competing interests.
