## [Peer Review File · Nature Communications]

REVIEWER COMMENTS

Reviewer #1 (Remarks to the Author):

In the paper „ Microtubule disassembly by caspases is the rate-limiting step of cell extrusion,“ the authors dissect the role of microtubules in the process of cell extrusion. As the process involves cell shape changes, which requires force, most studies focused on actomyosin-dependent contractile forces. Here, the authors provide new biological insights into the regulation of cell shape changes during extrusion by microtubules.

First, the authors establish that the observed actomyosin levels and localization are not required for cell shape changes associated with the early phase of cell extrusion. In addition, simulations indicate that actomyosin contractility would not lead to changes seen in extruding cells (Fig. 1, 2). Looking for the actomyosin independent mechanism, they tested microtubules, which were recently reported to generate protrusive/compressive forces to induce cell shape changes. This idea is consistent with the result showing that the microtubule cytoskeleton remodels at the onset of cell extrusion (Fig. 3). The authors demonstrate that the observed depletion of microtubules in extruding cells is regulated by caspases (Fig. 4). Next, they confirm that modulation of microtubule polymerization or depolymerization leads to changes in the cell apical area without affecting MyosinII activity (Fig. 5). Finally, they demonstrate that depolymerization of microtubules leads to cell extrusion, and thus by using nocodazole, they can bypass the requirement of caspase activation (Fig. 6).

Overall, this is an interesting paper, and the majority of conclusions seem well supported by the data. However, the authors need to consider the following comments before publication:

1) Although authors show that MyosinII levels do not change significantly during the initial phase of cell extrusion, it would be good to show that blocking MyosinII (for example, via injection of inhibitors into the notum - as they did with nocodazole) does not affect the initial rate of cell extrusion. If the model is correct, one would assume that the final stage would be blocked as actomyosin contractility is important for the deformation in later stages. As an alternative, the authors could quantify the initial change in the apical area (for example, before and 10 or 20 minutes after the onset of extrusion), and probe for the difference between wild-type cells and MyosinII inhibited cells.

2) Certainly, a more extensive characterization of microtubules is needed to interpret the mechanism of depletion and its importance:

a) In Figure 3B', the authors show the distribution of the microtubule signal along the A/B axis in one cell. To support this finding, it is crucial that the authors provide statistical significance (i.e. quantify more cells).

b) Figure S3I' also lacks statistics. The authors should compare the alpha-tubulin intensity at the level of adherens junctions at different time points (t=0,15min, 30 min, and 45min after the onset of extrusion).

c) The authors suggest that the observed reduction of the microtubule signal is due to microtubule disassembly. An alternative interpretation for microtubule reduction is the inhibition of microtubule nucleation/polymerization, which would also decrease the total microtubule number. One such example is the inactivation of the centrosome toward the end of mitosis (which causes the disappearance of centrosomal microtubules). The same could be true here. As the study deals with the novel function of microtubules during cell extrusion, it would be essential to show which mechanism (microtubule nucleation or depolymerization) is responsible for microtubule depletion. To distinguish between these

two mechanisms, the authors can quantify the number of microtubule comets during the extrusion process (EB1, as used in the paper). If the number of growing microtubules per cell stays the same, then the reduction in microtubule number should come from an increased disassembly rate. This could be validated by quantification of microtubule disassembly events (using GFP tagged tubulin) versus microtubule growth events (using EB1) per area (e.g., cell...) before and after the onset of extrusion.

d) As the depletion of microtubules can directly lead to cell extrusion, it would be good to show that stabilization of microtubules (for example, by Taxol) would inhibit cell extrusion. This additional experiment would unequivocally demonstrate that microtubules are required for extrusion.

3) The authors should discuss additional mechanisms that may be involved in destabilization/disassembly of microtubules (and regulated by caspases), such as inactivation of microtubule plus-end proteins and microtubule crosslinking. These mechanisms would allow microtubules to bear increased mechanical loads, and their inactivation would decrease their capacity to resist contractile forces and trigger extrusion.

Reviewer #2 (Remarks to the Author):

In this manuscript, Villars and the colleagues sought out the initiation of cell extrusion in the *Drosophila* pupal notum by using extensive *Drosophila* genetics, optogenetics, pharmaceutical approaches, biophysical perturbations, imaging, and quantitative analysis. This study found a surprising behavior of microtubules in cell extrusion: Microtubules disassemble at the beginning of cell extrusion and it initiates the apical constriction. This is a stark contrast to the conventional wisdom that the actomyosin contraction initiates the cell extrusion. The authors further demonstrated through rigorous experiments that a decrease (or an increase) of microtubules is sufficiently lead to a decrease (or increase) of cell apical area, which might be a general mechanism in tissue morphogenesis outside of cell extrusion. This study convincingly demonstrates that microtubules disassembly is a key rate-limiting step of extrusion. While some additional clarifications and/or works are necessary, I support the publication in *Nature Communications*.

Major comments.

1. This study investigates the role of caspase in microtubules disassembly by perturbing the caspase. In this current format, however, there is no clear demonstration that the caspase is activated in the cell that shows the microtubules disassembly. It would be more convincing if the authors image the caspase activity and the dynamics of microtubule simultaneously in non-perturbed tissue.

2. The authors elegantly used OptoDronc to disassemble microtubules and described "a rapid depletion of MTs upon activation of optoDronc in clones which was concomitant with cell apical constriction (Figure 4A,C)." However, Fig. 4C seems the apical constriction precedes microtubule disassembly. Could the authors 1) measure the time difference between the onset of perimeter change and the onset of SirTubulin intensity for each data, and 2) check which comes first? Alternatively, the authors could measure the correlation coefficient of the perimeter and the SirTubulin intensity.

Minor comments.

3. The following sentence starting from line 140 is not clear. "Accordingly... in good agreement with the dynamics we observed during the first 20 minutes of extrusion in the notum (Figure 2J)." Which data from the vertex model needs to be compared with the

experimental data shown in Fig. 2J?

4. The correlation coefficient shown in Fig. 3E indicates a 2min shift. If the data was taken every 5min, this 2min can be considered as "concomitant". If the time resolution of the data is 1min, then it is better to clarify which data precedes the others

5. One of the white arrows in Fig. 3G' indicates the onset of microtubule depletion. I feel that this arrow should be located at an earlier time. Probably close to the 40min mark and the time closer to the onset of apical contraction.

6. It would be helpful to explain in the text that "regions where no caspase activity is observed (line 245)" is outside of midline.

7. It would be helpful to explain in the text that "the depletion of Hid by RNAi (line 249)" leads to caspase inhibition.

8. I suggest explaining what "C3-" and "C3+" in Fig. 6E and 6E' represents or replacing them with "GC3Ai-" and "GC3Ai+".

9. It would be helpful to provide the WT data in Fig. 6H so that the line ", albeit not back to WT speed. (line 261)" is easy to follow.

Reviewer #3 (Remarks to the Author):

Review of manuscript by Villars et al., entitled 'Microtubule disassembly by caspases is the rate-limiting step of cell extrusion'

In this study Villars et al. quantitatively analyse the dependency of apoptotic cell extrusion in the developing *Drosophila* larval notum. Cell extrusion events are critically important for epithelial homeostasis as well as in some cases epithelial morphogenesis, and previous studies have always shown key roles for (apical) actomyosin within the dying extruding cells. Here the authors show that in the fly notum the initial apical constriction observed does not involve actomyosin or depend on it but instead requires the caspase-dependent disassembly of an apical-medial microtubule meshwork.

This is a very interesting study, emphasizing yet again the so far usually underestimated role for microtubules in morphogenetic and homeostatic events.

Unfortunately, though, I find there are quite a few limitations in how clearly the data shown in the figures support the conclusions drawn by the authors, and I feel the data need strengthening before any publication.

I explain in detail below where I have found the data presented not clear, not as supportive as interpreted, or where alternative interpretation are completely valid but were not excluded:

1) Figure 1: the authors state in line 94-95 that a supracellular actomyosin cable forms. I

assume what they want to refer to is a cable in all the cells surrounding the extruding one? I cannot see that in any of their images. Has this been demonstrated for the notum cell extrusion? From the images they show it is impossible (without some sort of clonal labelling analysis) to decide in which cell the actin and myosin accumulations are, the central extruding one or the surrounding neighbours. Please clarify, supply more data or rephrase.

I am somewhat confused by their definition of 'onset' as is labelled in the figures. They say in the legend to Figure 1 that ' $t=0$ is the time of extrusion termination (no apical area visible).' But then they also say that the 'grey dotted line represents the onset of extrusion marked by the inflection of the perimeter curve'. Curiously though the grey line is at $t=0$, meaning extrusions take no time, so this cannot be correct.

I am still also not convinced that just not seeing an increase in junctional plus apical-medial myosinII necessarily proves that it is not involved early on in the extrusion, when there very clearly is a lot of myosin around, especially in interesting junctional and peri-junctional arrangements already from the start. The best test would be to disrupt myosin or its activators to test for requirement.

2) Figure 2:

Lines 127 onwards discussing the modelling:

The purse-string cable that could form to help extrusion would be a feature of the surrounding cells, so would modelling an increase in line tension for the central cell really capture that?

3) Figure 3 C: from the still images I can see that the central extruding cell is showing a lowering of JupiterGFP levels, but equally, there are lots of cells just above that also show low MT levels, and other cells below that also show a lowering, but these are not apoptotic. Similarly, in Movie S8, the cell just above the extruding one also seems to lower MTs apically but that does not affect its apical area... How does this fit with the later experiments where the authors aim to show sufficiency of microtubule levels for increase or decrease of apical area. What effect on apical area do fluctuations of microtubule levels in the wild-type have? Either these changes in MT levels have an effect all the time, even in wt conditions where levels appear to fluctuate, or not.

For movies S8,9,10 it would be ideal to see the MT-related channel also by itself, please!

Another issue: the authors use overexpression of UAS-Shot-L(C)-GFP to visualise Shot (shown in Figure S3E). This isoform C lacks the actin-binding calponin homology domain (which the authors do not comment on at all here) and has been reported to bundle microtubules (Booth et al., DevCell 2014), so it does not report on wild-type Shot localisation and in fact changes microtubule behaviour when overexpressed. The best life Shot label is a GFP-exon trap that various labs hold (originally from Kyoto stock centre w[1118]; PBac{EGFP-IV}shot[KM0008] (DGRC number 109707), this labels one of the longest isoforms with all important domains. Or if a UAS construct is required, then the L(A) isoform also available from Bloomington contains both the actin-binding and microtubule-binding domains and at least in embryos and the germline looks most like anti Shot antibody labelling.

So the statement in lines 168-170 is not correct as UAS-Shot-L(C) will not report on endogenous Shot localisation.

A general question with regards to the changes in microtubules reported:

Have the authors analysed microtubules in the whole volume of the cell? I .e. how certain are they it is really a reduction rather than a reorganisation as reported in other apically

constricting cells (Booth et al., DevCell 2014)? The authors show that centrosomes remain apical, but does the association of microtubules with centrosomes change? Are microtubules nucleated at centrosomes in these cells? If so, why are not more microtubules constantly being nucleated, what changes? Do centrosomes change?

The authors show α -tubulin-mCherry in Figure S3 H to illustrate a reduction of MTs from apical to basal in three optical sections. This label though does not seem to allow visualisation of microtubule filaments or bundles, it is just a hazy label. And whereas I can appreciate a reduction in intensity in the apical section, the lateral and basal section do not show clearly what is quantified in I, I'. In fact, even within the apical region there seem to be several other cells whose apical level of α -tubulin-mCherry is decreasing over the time-points shown, even though these cells are not dying or extruding. So how do we know that there are not just general fluctuation of apical MT intensity occurring? A better microtubule label should be used for the whole cell analysis, like SiR-tubulin or Jupiter-GFP.

4) Figure S4, A-B:

The authors test here whether the constriction of the apical area could be the inducer of the observed microtubule disassembly. From the images shown, I cannot tell which cell they are referring to and therefore I cannot tell if the images support their statement. The E-Cadherin label, especially for the highlighted section, is very hazy and does not clearly label cell outlines, it is also not shown in magnification. Furthermore, it is impossible for the reader to tell if the laser cutting led to any tissue relaxation, change in apical area, as no comparison of apical areas pre-and post-cut (in a way where individual cells can be assessed) is shown. This should be rectified to make the statement that apical constriction as a driver of MT disassembly can be excluded.

Figure 4:

Panels in A: the dotted line outlining the clone does not seem to match where the magenta label of the optoDronc can be seen. Again, various other cells also seem to show fluctuations of the sirTubulin label, so how do we know it is specific?

Panels in D: Again, also other cells show random much higher intensity of the sirTubulin, and of the two outlined cells only one shows the increase, so how specific is this?

Panels in G: Again, there are many not small cells inside and outside of the clone that show equally high intensity of acetylated α -tubulin, so how can the reader assess that this is not random fluctuations of apical MT intensity levels?

5) Figure 5:

Judging from panel A'' (before injection) the apical microtubules usually span a lot of the apical area. The EB1-GFP labelled microtubules that have regrown after 125 sec seem to still be very short as shown in panel D. How do the authors think these microtubules are going to affect the apical cell area? What mechanism are they supposing would be at work to increase the area? I can see the argument that removing MTs might be a requirement for apical constriction to occur, as the microtubules might otherwise sterically block the process, but I cannot think of a mechanism that would allow microtubules to easily increase a cell's apical area, especially not just newly nucleated ones that are still very short.

With regards to the LARIAT experiment: as the authors use overexpression of tagged α -tubulin, to they have any way of assessing what happens to endogenous tubulins present in these cells that most likely are all not affected? To be able to correctly interpret this experiment the authors need to know and show what happens to endogenous tubulin and microtubules overall. If they inject sirTubulin into these pupae, are apical microtubules actually disappearing in the LARIAT expressing and activated cells? This is essential to show!

Also, the change in perimeter observed, though at one timepoint significant statistically, seems very small indeed. How can the authors exclude that it is not the effect of massive

aggregates within the cells and some response to that that leads to the very mild change observed?

6) Figure 6:

The authors state that spastin-activation in clones appears to increase extrusions. There seems to be a lot of variability, though, from pupa to pupa, in terms of the number of extrusions. The overall numbers in between control and experiment in A seems quite different, and even more so, there seems to be a huge difference in the number of extrusions between the controls in A and in C. Analysed were 3 pupae with spastin clones and 2 pupae as control. With such differences in numbers, at least as it appears from the images shown, more pupae need to be analysed.

Also, with regards to spastin-expression: in our and other labs' hands the microtubule depletion induced by spastin overexpression in different tissues tends to not be complete, in the sense that not all cells that express spastin will show a decrease in microtubules (and it is unclear why that is, but it has been observed by several labs). So have the authors analysed how effective the clonal expression of spastin is on microtubules? Have they stained for α -tubulin and acetylated α -tubulin upon spastin induction? This should be shown.

If colcemid injection, even in the wt, increases extrusions across the notum, as the authors seem to show in this figure, then interpreting what they show in Figure 5 with colcemid injections is much harder as there must be so much unusual extrusions going on. So it is hard to know how individual cells will respond in a tissue with such changes. This effect of colcemid should have been mentioned in the discussion of Figure 5.

A more general question: by what mechanism do the authors assume does a depletion of microtubules lead to cell extrusion? So far the authors were arguing that microtubule depletion is permissive for apical area reduction and extrusion in the wild-type, but with Figure 6 they seem to argue that it is sufficient to drive it. How? What happens once microtubules are gone? Because this is clearly not what happens in many other tissues, including the embryonic epidermis at various stages, when spastin is overexpressed here. So what is different about notum cells, what is the (presumed, hypothesised) molecular mechanism?

With regards to figure panels in this figure:

The Cadherin cell outline label shown in E and E' is really not very good, why is this? It is impossible for the reader to see the cell outlines, so impossible to judge how accurately the dotted line represents them.

Also, what is happening to the cells surrounding the extruding cell in panels in G? They seem to massively increase their apical area judged by the much better E-Cad label shown here. What is going on?

7) Discussion and title:

The authors state at the beginning of the discussion: 'This is, to our knowledge, one of the first descriptions of a role of MTs in the initiation of cell extrusion independently of MyoII.' This is a somewhat confusing way of phrasing the results. MTs do not have 'a role...in the initiation..', rather, what the authors aim to show is that they need to be gone for extrusion to happen. This phrasing somehow implies an active role of MTs, whereas what the authors argue is that their disassembly is permissive.

With regards to the title 'Microtubule disassembly by caspases is the rate-limiting step of cell extrusion', I think it would be more correct to say 'Microtubule disassembly by caspases is a rate-limiting step of cell extrusion'. The authors have not analysed or shown here whether the microtubule disassembly is the only rate-limiting step.

Answer to reviewers

“Microtubule disassembly by caspases is an important rate-limiting step of cell extrusion”

General comments

We would like to thank all the reviewers for their thoughtful comments and suggestions which helped to significantly improve the quality of this manuscript. While they all valued the importance of our findings, they brought up some concerns that we can summarise in three main categories:

1. The role of MyoII in cell extrusion.

We showed the lack of correlation between MyoII changes/dynamics/concentration and the onset of constriction during extrusion. This led to the suggestion that other components must also contribute to the initiation of cell extrusion. Importantly, we did not claim that actomyosin contractility has no function in this process but rather that this is not sufficient to explain the dynamics observed at the onset of extrusion. We also found that actomyosin accumulates at the late phase of extrusion (last 10 min) similar to previously described actomyosin rings. Reviewer 1 and 3 asked for a more detailed characterisation of the contribution of MyoII to extrusion termination. While we purposely didn't focus our attention on the ring formation (which has been extensively studied in MDCK cells[1-3], or in the pupal abdomen[4]), we provide now a better description of this late phase by combining two-colour labelling of MyoII and sorting the contribution of the dying and the neighbouring cell. In agreement with previous studies[2-4], we first observed an accumulation in the dying cell and then in the neighbours (new **Figure S1B**). We also tested the functional impact of the impairment of MyoII activation. Depletion of the main activator of MRCL (using dsRNA against ROCK) significantly impact the epithelial architecture and blocked cytokinesis, but did not prevent cell extrusion nor affected significantly the duration of cell extrusion (new **Figure S2B,C**). This suggests that cell extrusion is poorly sensitive to global changes of actomyosin activity, in agreement with previous work on neuroblast delamination[5], and our correlative analysis. Yet, we noticed that the regime of cell extrusion becomes slightly different with a strong decrease of circularity, and the absence of late cell rounding that we observed in the WT situation (which correlates with the formation of the actomyosin ring). This is in agreement with a regime purely driven by a reduction of cell resting area, as observed in our vertex model. This also illustrates the great plasticity and robustness of extrusion which can use different mechanical strategy to exclude the cell. Importantly, while ROCK inhibition had a limited impact on the speed of cell extrusion, stabilisation of microtubules by Taxol lengthens significantly the extrusion duration (new results, **Figure 6I-K**). Altogether, this argues for MT depletion being an important rate limiting step of extrusion rather than the upregulation of actomyosin in the pupal notum.

2. Relationship between caspase activation and microtubule depletion

We showed that microtubules depletion occurs downstream of effector caspase activation, however, as noted by reviewer 2, we have not documented the relative timing of caspase activation and MTs depletion in WT conditions. Yet, we were confident that caspase activation did precede MT depletion as we previously showed that caspase activation systematically preceded the onset of cell extrusion[6-8], while we showed in this study that the onset of extrusion is concomitant with the depletion of MTs (**Figure 3**). Using GC3Ai (a live effector caspase sensor), we now confirmed that caspases are activated in cells depleting microtubules (new **Figure S4D**).

3. Mechanism of depletion of microtubules during cell extrusion

Several reviewers asked for a more detailed characterisation of the mechanism of MT depletion (reviewer 1 and 3). These included :

- a better characterisation of MT depletion throughout the cells and excluding an alternative mechanism based on MT relocalisation
- data regarding the dynamics of polymerisation/depolymerisation rate of MTs during extrusion
- the significance of the depletion of MTs compared to the spontaneous fluctuations
- the lack of some controls (LARIAT and Spastin experiments)
- Confirming the role of MTs depletion in extrusion by promoting their stabilisation

We have now addressed all these concerns by:

- Documenting MT depletion with different markers throughout the cell in 3D during spontaneous extrusion and upon activation of caspases with optoDronc, and confirming the absence of association between apical MTs and the centrosome.
- Analysing the relative dynamics of MT +end dynamics (EB1-GFP) and total MT (sirTub) during the depletion
- Better documenting the spontaneous fluctuations of MTs in control cells relative to the systematic depletion observed during extrusion
- Providing further controls for the LARIAT and Spastin experiments
- Confirming the essential role of MT depletion during extrusion by characterising the very significant slow-down of extrusion upon taxol injection (which is not observed upon ROCK depletion).

Altogether, these results confirmed that MT depletion occurs throughout the cells and that the dynamics is compatible with a depletion of monomers which would affect both polymerisation and depolymerisation rates. More importantly, we confirm that MT depletion is an important rate limiting step of extrusion since their stabilisation by Taxol is sufficient to drastically slow-down the speed of extrusion (while ROCK depletion does not).

Point by point answer to reviewers

Reviewer comments in "Calibri"

Answer in "Arial"

Reviewer #1 (Remarks to the Author):

In the paper „ Microtubule disassembly by caspases is the rate-limiting step of cell extrusion,“ the authors dissect the role of microtubules in the process of cell extrusion. As the process involves cell shape changes, which requires force, most studies focused on actomyosin-dependent contractile forces. Here, the authors provide new biological insights into the regulation of cell shape changes during extrusion by microtubules.

First, the authors establish that the observed actomyosin levels and localization are not required for cell shape changes associated with the early phase of cell extrusion. In addition, simulations indicate that actomyosin contractility would not lead to changes seen in extruding cells (Fig. 1, 2). Looking for the actomyosin independent mechanism, they tested microtubules, which were recently reported to generate protrusive/compressive forces to induce cell shape changes. This idea is consistent with the result showing that the microtubule cytoskeleton remodels at the onset of cell extrusion (Fig. 3). The authors demonstrate that the observed depletion of microtubules in extruding cells is regulated by

caspsases (Fig. 4). Next, they confirm that modulation of microtubule polymerization or depolymerization leads to changes in the cell apical area without affecting MyosinII activity (Fig. 5). Finally, they demonstrate that depolymerization of microtubules leads to cell extrusion, and thus by using nocodazole, they can bypass the requirement of caspase activation (Fig. 6).

Overall, this is an interesting paper, and the majority of conclusions seem well supported by the data. However, the authors need to consider the following comments before publication:

Answer: We thank the reviewer for the positive comments and the thoughtful suggestions. We believe we have addressed all the concerns, which now really strengthen the role of MT in cell extrusion (specially by comparing the impact of ROCK depletion compared to Taxol injection).

1) Although authors show that MyosinII levels do not change significantly during the initial phase of cell extrusion, it would be good to show that blocking MyosinII (for example, via injection of inhibitors into the notum - as they did with nocodazole) does not affect the initial rate of cell extrusion. If the model is correct, one would assume that the final stage would be blocked as actomyosin contractility is important for the deformation in later stages. As an alternative, the authors could quantify the initial change in the apical area (for example, before and 10 or 20 minutes after the onset of extrusion), and probe for the difference between wild-type cells and MyosinII inhibited cells.

Answer: We thank the reviewer for this very relevant suggestion. As suggested, we have depleted ROCK by RNAi throughout the pupal notum and measured the impact on cell extrusion. The impact of ROCK depletion was clearly visible based on the altered morphology of the tissue and the presence of multinucleated cells (due to cytokinesis failure). Of note, there was still some actomyosin located at the junction and this did not completely abolish Myosin recruitment. Interestingly, in that context, we do not see any changes in the initial rate of constriction nor in the total duration of extrusion. This is coherent with the expectation of the reviewer and again argues for a predominant role of microtubules depletion during the initial phase of contraction. Interestingly, we did not completely block the termination of extrusion, but rather observed a change of the regime: while WT extrusion is associated with an increase of rounding (as expected from the increase of line tension driven by the actomyosin ring), cell circularity keeps decreasing and reaches much smaller value in ROCK RNAi (**new Figure S2B,C**), suggesting that the cell expulsion is now completely dominated by the decrease of cell resting area (as observed in our vertex model, **Figure 2**). These results suggest that cell extrusion is very robust to global changes in actomyosin activity (as previously observed in neuroblast delamination in the fly embryo[5]) and illustrate the plasticity of the process. These results also must be compared with the very significant impact of microtubule stabilisation on the duration of cell extrusion, which argue again for a predominant role of MTs depletion in the regulation of extrusion timing/duration.

We now described these new results in the main text as followed (line 152) :

*“Importantly, while depleting ROCK (the main activator of MyoII[9]) by RNAi had a stringent effect on the epithelial morphology and blocked cytokinesis, it had no significant impact on the speed of extrusion (**Figure S2B,C**). In this context, the deformations occurring during extrusion were now associated with a strong decrease of circularity and the absence of cell rounding in the late phase (**Figure S2C**), confirming the link between the late rounding observed in WT extrusion and the increase of actomyosin (**Figure 2J**). Thus, global modulation of MyoII activation has little impact on cell extrusion in the notum, in agreement with an extrusion regime poorly relying on changes in line tension.”*

2) Certainly, a more extensive characterization of microtubules is needed to interpret the mechanism of depletion and its importance:

- a) In Figure 3B', the authors show the distribution of the microtubule signal along the A/B axis in one cell. To support this finding, it is crucial that the authors provide statistical significance (i.e. quantify more cells).

Answer: We thank the reviewer for this suggestion. This figure was mostly there to document the distribution of MTs in control cells and show the strong pool in the medio-apical plane. We have confirmed this profile by adding more cells (new **Figure 3B'**). Of note, we observed a similar apical pool with all the sensor that we used (Jupiter, EB1-GFP, sir-tub, α tub-mCherry, **Figure 3** and **Figure S3-1**). We also now provide lateral views with other markers to document the distribution along the apico-basal axis in single cells expressing EB1-GFP (new **Figure S3-2 C,D**). We have also better documented the orientation of MTs by tracking EB1-GFP comets (new **Figure S3-1 A,A'**). This clearly showed that these medio-apical MTs are not emanating from a unique point (the centrosome) in agreement with a pool mostly composed of non-centrosomal apical MTs.

- b) Figure S3I' also lacks statistics. The authors should compare the alpha-tubulin intensity at the level of adherens junctions at different time points (t=0,15min, 30 min, and 45min after the onset of extrusion).

Answer: We thank the reviewer for this comment. We would like to stress the fact that 3D characterisation of MTs dynamics is tedious in this tissue where cells can have very convoluted morphology on the basal sides (where the contribution of the signal from the dying cells and its neighbours is hard to sort). As a result, we can get fairly variable results when performing the quantification. To better document the depletion throughout the z-axis, we now confirmed this dynamics by providing statistical test for the variation of α tubulin-mCherry (new **Figure S3-2 B'**). To better document the variations along z-axis we now provide a lateral view of a single cell expressing EB1-GFP during extrusion which confirm a global depletion of MTs (new **Figure S3-2 C,D**). More importantly, we unambiguously observed a depletion of MTs throughout the cell upon activation of optoDronc in clones both on the apical and basal side (new **Figure S3-2 E,E'**), which is not compatible with a model based on a specific depletion of the apical pool or a relocalisation of MTs. Based on these new results, we are confident that MTs depletion is indeed taking place throughout the cell during cell extrusion both on the apical and basal sides.

- c) The authors suggest that the observed reduction of the microtubule signal is due to microtubule disassembly. An alternative interpretation for microtubule reduction is the inhibition of microtubule nucleation/polymerization, which would also decrease the total microtubule number. One such example is the inactivation of the centrosome toward the end of mitosis (which causes the disappearance of centrosomal microtubules). The same could be true here. As the study deals with the novel function of microtubules during cell extrusion, it would be essential to show which mechanism (microtubule nucleation or depolymerization) is responsible for microtubule depletion. To distinguish between these two mechanisms, the authors can quantify the number of microtubule comets during the extrusion process (EB1, as used in the paper). If the number of growing microtubules per cell stays the same, then the reduction in microtubule number should

come from an increased disassembly rate. This could be validated by quantification of microtubule disassembly events (using GFP tagged tubulin) versus microtubule growth events (using EB1) per area (e.g., cell...) before and after the onset of extrusion.

Answer: Thank you for raising this very interesting point. First, we would like to stress the fact that we have limited spatial and temporal resolution in our system (mostly due to the size of cells, the relative low signal to noise, but also to the non-predictable nature of when and where an extrusion will take place). It is therefore not really possible to get a single MT resolution. We can however track single EB1 comets and also compare the relative dynamics of EB1 and total MT pool (using sirTubulin) during extrusion. Using higher temporal resolution (every 30 sec, the best trade-off to detect extruding cells and image several z-planes), we checked for potential delays between the modulation of MT total pool and polymerisation rate with EB1-GFP. Using cross-correlation, we could not find any significant delay between the two markers, suggesting that both polymerisation and depolymerisation rates are affected concomitantly (at least with our limited time and spatial resolution). This is in good agreement with a model where α and/or β -tubulin would be cleaved by caspases (as both show cleavage sites for caspases) hence progressively depleting monomers and affecting both polymerization and depolymerization rates. These new results are described in new Figure **S3-1 I-J** And in the results section, line 202:

“We then checked whether this depletion was dominated by a change of polymerisation rate or depolymerisation rate of MTs by assessing putative lag-time between the change of EB1 comets (new growing MTs) and changes in the total pool of MTs (using sirTubulin). We could not detect any significant lag-time between the reduction of EB1 and total pool of MTs during extrusion (Figure S3-1 I-J), which could be compatible with a process affecting both the polymerisation and the depolymerisation rate (see below and Discussion).”

We also include a paragraph in the discussion which also mention the limitation of these analysis due to our limited spatial and temporal resolution and open the possibility for more subtle differences between polymerisation and depolymerisation dynamics, line 374:

“The disassembly of MTs could either be driven by a reduction of the polymerisation rate, an increase of the depolymerisation rate or both. In principle, this could be sorted by comparing the number of EB1 positive comets relative to the total number of MTs over time. However, we found that EB1-GFP signal and total MT signal (using sirTubulin) decrease concomitantly during extrusion with no significant lag-time (Figure S3-1 I-J). While we cannot exclude that our limited temporal and spatial resolution may miss subtle time differences, this is compatible with a mechanism affecting both the polymerisation and the depolymerisation rate, as would happen upon sequestration/destruction of Tubulin monomers.”

d) As the depletion of microtubules can directly lead to cell extrusion, it would be good to show that stabilization of microtubules (for example, by Taxol) would inhibit cell extrusion. This additional experiment would unequivocally demonstrate that microtubules are required for extrusion.

Answer: This is indeed a very important point. We actually previously tried to inject Taxol but failed to obtain a significant impact on MTs (no visible effect on MT dynamics or concentration) and reached the limit of solubility of Taxol in Ethanol. To increase the concentration of Taxol, we have now performed the same experiment with Taxol diluted in DMSO. To reach high concentrations, we had to reach the limit of Taxol solubility in DMSO (60 mM) and used pure DMSO injection as a control. While this had an effect on the pupal notum (roughly 6-8h after injection, the tissue starts to collapse), Taxol injection led to a rapid effect on MTs dynamics and concentration, and inhibited cytokinesis (1 hour after

injection) which was not observed in DMSO injected pupae. We used this time window to analyse the impact on extrusion dynamics. This led to a very significant increase of cell extrusion duration (~30 min in DMSO control, ~150 min in Taxol) ranging from relatively subtle slow-down to drastic reduction of the constriction rate (**new Figure 6 I-K, movie S21**). Moreover, these slow constrictions are associated with a significant cell rounding, very similar to the profile observed in optoDronc cells combined with p35 (**Figure 4G**), in agreement with a slow constriction associated with a progressive increase of line tension (**Figure 2**). Altogether, this experiment strongly argues for MTs depletion being an important rate limiting step of extrusion. We would like to thank again the reviewer to encourage us to test further this direction which was proven to be very instructive. These results are mentioned at line 310:

*“To confirm that MTs depletion is indeed an important rate-limiting step of extrusion, we stabilised MTs using Taxol injection at high concentration. While it did not totally block cell extrusion, it led to a global and drastic slowed-down of the speed of extrusion with variable durations (> 3 folds increase of extrusion duration, **Figure 6I-K, movie S21**). Moreover, the regime of deformation was now associated with a constant and significant increase of circularity (**Figure 6J**), similar to the deformations observed upon increase of line tension (**Figure 2G**), or upon optoDronc activation combined with p35 expression (**Figure 4H**).”*

3) The authors should discuss additional mechanisms that may be involved in destabilization/disassembly of microtubules (and regulated by caspases), such as inactivation of microtubule plus-end proteins and microtubule crosslinking. These mechanisms would allow microtubules to bear increased mechanical loads, and their inactivation would decrease their capacity to resist contractile forces and trigger extrusion.

Answer: As discussed above, these alternative mechanisms are now described in the discussion while mentioning the limit of the temporal and spatial resolution we can have in our system, including also line 391:

“Moreover, we cannot exclude at this stage alternative mechanisms of MTs destabilisation based on the modulation of plus-end binding proteins or crosslinkers by caspases.”

Reviewer #2 (Remarks to the Author):

In this manuscript, Villars and the colleagues sought out the initiation of cell extrusion in the *Drosophila* pupal notum by using extensive *Drosophila* genetics, optogenetics, pharmaceutical approaches, biophysical perturbations, imaging, and quantitative analysis. This study found a surprising behavior of microtubules in cell extrusion: Microtubules disassemble at the beginning of cell extrusion and it initiates the apical constriction. This is a stark contrast to the conventional wisdom that the actomyosin contraction initiates the cell extrusion. The authors further demonstrated through rigorous experiments that a decrease (or an increase) of microtubules is sufficiently lead to a decrease (or increase) of cell apical area, which might be a general mechanism in tissue morphogenesis outside of cell extrusion. This study convincingly demonstrates that microtubules disassembly is a key rate-limiting step of extrusion. While some additional clarifications and/or works are necessary, I support the publication in Nature Communications.

Answer: We thank the reviewer for the very positive comments about our work and the constructive suggestions.

Major comments.

1. This study investigates the role of caspase in microtubules disassembly by perturbing the caspase. In this current format, however, there is no clear demonstration that the caspase is activated in the cell that shows the microtubules disassembly. It would be more convincing if the authors image the caspase activity and the dynamics of microtubule simultaneously in non-perturbed tissue.

Answer: This is indeed a very relevant point. First, we would like to stress the fact that our previous published work demonstrated that effector caspase activation always precedes the onset of cell extrusion in the pupal notum[6-8]. In this study, we showed that MTs depletion is concomitant with the onset of cell extrusion. We were therefore quite confident that caspase activation would systematically precede MTs depletion. However, as it is a key point of our manuscript, we further document this statement by imaging pupae co-expressing UAS-GC3A1 (an effector caspase activity reporter) together with UAS- α -Tubulin-mCherry. Doing so, we could confirm that effector caspase were systematically activated prior to MTs depletion (**new Figure S4D, movie S13**).

This data is mentioned at line 225:

“In agreement with caspase activation systematically preceding extrusion and MT depletion being concomitant with the onset of extrusion (this study), effector caspase activation (using the live marker GC3A1[10]) was systemically observed in cells depleting MTs and extruding (Figure S4D, movie S13). We then checked whether caspase activation was sufficient to deplete MTs. “

2. The authors elegantly used OptoDronc to disassemble microtubules and described “a rapid depletion of MTs upon activation of optoDronc in clones which was concomitant with cell apical constriction (Figure 4A,C).” However, Fig. 4C seems the apical constriction precedes microtubule disassembly. Could the authors 1) measure the time difference between the onset of perimeter change and the onset of SirTubulin intensity for each data, and 2) check which comes first? Alternatively, the authors could measure the correlation coefficient of the perimeter and the SirTubulin intensity.

Answer: Thanks for pointing this issue. This apparent shift was actually related to the small number of cells that we quantified in this condition. We restricted this analysis to single clonal cell expressing optoDronc (the activation of Dronc in a group of neighbouring cells can affect the extrusion process and may change the speed[8]), which is relatively rare for clone induced more than 48h before imaging. We sampled more single optoDronc cells and provide now a curve based on more cells (where no apparent shift is visible) (new **Figure 4C,D**). We also calculated the cross-correlation between the cell perimeter and sirTubulin intensity in optoDronc cells and could not find a significant delay between the two, as observed in spontaneous extrusion with Jupiter-GFP, sir-Tubulin and EB1-GFP (**Figure 3E, new Figure S3-1 J**).

This new data is mentioned at line 231:

“We observed a rapid depletion of MTs (visualised with injected sirTubulin) upon activation of optoDronc in clones which was concomitant with cell apical constriction (Figure 4B-E, movie S14 top, no significant lag-time between sirTub diminution and perimeter constriction).”

Minor comments.

3. The following sentence starting from line 140 is not clear. "Accordingly... in good agreement with the dynamics we observed during the first 20 minutes of extrusion in the notum (Figure 2J)." Which data from the vertex model needs to be compared with the experimental data shown in Fig. 2J?

Answer: We apologise if this was not very clear. We were referring to the decrease of circularity that was observed in the experimental data for the first 20min of extrusion compared to trend observed in the Vertex model in **Figure 2H** for all the set of parameters reducing resting area (different rate of reduction of resting area, we always see a decrease of cell circularity). We have modified the main text to make a clearer reference to the figure line 151:

*“Accordingly, the reduction of the resting area of a single cell in the vertex model led to cell constriction concomitant with a progressive reduction of cell roundness (**Figure 2A,D,F,H, movie S5**), in good agreement with the dynamics we observed during the first 20 minutes of extrusion in the notum (**Figure 2J, compare to yellow to purple lines in Figure 2H with different rate of reduction of resting area**).”*

4. The correlation coefficient shown in Fig. 3E indicates a 2min shift. If the data was taken every 5min, this 2min can be considered as “concomitant”. If the time resolution of the data is 1min, then it is better to clarify which data precedes the others

Answer: We thank the reviewer for pointing this out. Actually this piece of data was obtained with a framerate of one minute (we now clearly indicate this in the legend). To check if this apparent shift is significant, we performed a statistic test comparing the delta t observed on each single cross-correlation curve (one per cell) to a theoretical value equal to 0 (no lag time). Since this test was not significant ($p=0.73$) we concluded that there is no significant delay between Jupiter-GFP depletion and perimeter contraction onset. The result of the test is now clearly indicated on the **Figure 3E**.

5. One of the white arrows in Fig. 3G' indicates the onset of microtubule depletion. I feel that this arrow should be located at an earlier time. Probably close to the 40min mark and the time closer to the onset of apical contraction.

Answer: This is correct. The arrows we placed manually to guide the readers but it is true that the apical area starts shifting at 40min. We have moved the arrow accordingly.

6. It would be helpful to explain in the text that “regions where no caspase activity is observed (line 245)” is outside of midline.

Answer: We apologise for this lack of explanation. We now clearly state in the text that this is outside the midline and posterior regions. Line 288:

*“Conditional induction of Spastin (a MT severing protein[11]) in clones **was sufficient to deplete MTs (Figure S6A) and increase the rate of extrusion, including outside the midline in regions where no caspase activity is observed and where very few cells die in control conditions[6-8] (Figure 6A-B, movie S17).**”*

7. It would be helpful to explain in the text that “the depletion of Hid by RNAi (line 249)” leads to caspase inhibition.

Answer: We apologise again for the lack of explanation. We now clearly explain that hid is a central pro-apoptotic gene essential for caspase activation and cell extrusion in the pupal notum. Line 293:

“We used the inhibition of the pro-apoptotic gene hid which blocks caspase activation and cell extrusion in the notum[7].”

8. I suggest explaining what “C3-“ and “C3+“ in Fig. 6E and 6E' represents or replacing them with “GC3Ai-“ and “GC3Ai+“.

Answer: We have followed your suggestion and changed **Figure 6** accordingly.

9. It would be helpful to provide the WT data in Fig. 6H so that the line “, albeit not back to WT speed. (line 261)” is easy to follow.

Answer: This is a good suggestion. We now included the data from the control optoDronc in the same graph.

Reviewer #3 (Remarks to the Author):

Review of manuscript by Villars et al., entitled ‘Microtubule disassembly by caspases is the rate-limiting step of cell extrusion’

In this study Villars et al. quantitatively analyse the dependency of apoptotic cell extrusion in the developing *Drosophila* larval notum. Cell extrusion events are critically important for epithelial homeostasis as well as in some cases epithelial morphogenesis, and previous studies have always shown key roles for (apical) actomyosin within the dying extruding cells. Here the authors show that in the fly notum the initial apical constriction observed does not involve actomyosin or depend on it but instead requires the caspase-dependent disassembly of an apical-medial microtubule meshwork.

This is a very interesting study, emphasizing yet again the so far usually underestimated role for microtubules in morphogenetic and homeostatic events.

Unfortunately, though, I find there are quite a few limitations in how clearly the data shown in the figures support the conclusions drawn by the authors, and I feel the data need strengthening before any publication.

I explain in detail below where I have found the data presented not clear, not as supportive as interpreted, or where alternative interpretation are completely valid but were not excluded:

Answer: We thank the reviewer for the positive comments about our manuscript and acknowledging the underestimated role of microtubules in morphogenesis. We also thank the reviewer for the suggestions that helped us to improve significantly the quality of the data and our demonstration.

1) Figure 1: the authors state in line 94-95 that a supracellular actomyosin cable forms. I assume what they want to refer to is a cable in all the cells surrounding the extruding one? I cannot see that in any of their images. Has this been demonstrated for the notum cell extrusion? From the images they show it is impossible (without some sort of clonal labelling analysis) to decide in which cell the actin and myosin accumulations are, the central extruding one or the surrounding neighbours. Please clarify, supply more data or rephrase.

Answer: It is true that we assumed that the cable of actomyosin was supracellular and intracellular based on previous publications mostly from cell culture[1-3] but also in the *Drosophila* pupal abdomen [4]. We voluntarily didn't perform a thorough characterisation as we decided to focus on the early phase of constriction where we could not see changes in actomyosin. Still, as suggested by the referee we have used a two colour system to visualise the contribution of MyoII from the dying cells and its neighbours (using mitotic recombination with FRT40A MyoII-GFP and FRT40A MyoII-RFP). We could confirm that the actomyosin ring could be decomposed in a cell autonomous contribution followed by the formation of a supracellular cable in the neighbours (new **Figure S1B**). We mention now this result in line 99:

“Similar to extrusion in other systems[1-4], both the dying cells and its neighbours contribute to the late accumulation of MyoII, the cell autonomous accumulation slightly preceding the accumulation in the neighbours (Figure S1B).”

I am somewhat confused by their definition of ‘onset’ as is labelled in the figures. They say in the legend to Figure 1 that ‘t=0 is the time of extrusion termination (no apical area visible).’ But then they also say that the ‘grey dotted line represents the onset of extrusion marked by the inflection of the perimeter curve’. Curiously though the grey line is at t=0, meaning extrusions take no time, so this cannot be correct.

Answer: We apologise if this was confusing, we indeed sometimes defined t0 as the onset of constriction while in other figures we defined t0 as the termination of apical constriction. We have now consistently use t0 as the termination of extrusion, except in the context where extrusion is blocked (p35 + optoDronc) where we have many cells which do not terminate extrusion on the time scale of the movie (here t0 is set at the beginning of the movie).

I am still also not convinced that just not seeing an increase in junctional plus apical-medial myosinII necessarily proves that it is not involved early on in the extrusion, when there very clearly is a lot of myosin around, especially in interesting junctional and peri-junctional arrangements already from the start. The best test would be to disrupt myosin or its activators to test for requirement.

Answer: We thank the reviewer for giving us a chance to clarify this point. First, we would like to stress that we did not claim that MyoII is not necessary for the constriction (most likely a minimal level of contractility is required to undergo extrusion), but rather that this is not sufficient to explain the dynamics of extrusion. This is not only supported by our thorough characterisation of actomyosin dynamics (including the change of constriction yield) but also by the strong MyoII accumulation we observed in optoDronc cells combined with p35 (which do not extrude, or very slowly) and the geometrical profile of cell deformation (which is not compatible with a constriction driven by an increase of line tension, **Figure 2**). Yet, since this is a very important point, and also as suggested by Reviewer 1, we have now characterised extrusion upon depletion of ROCK by RNAi in the notum. The reduction of MyoII activity was clearly indicated by the modified architecture of the tissue and the numerous cytokinesis failure and presence of polynucleated cells. Yet, this condition did not abolish cell extrusion, nor affected the averaged duration of extrusion (**new Figure S2B,C**). It does however change the regime of cell deformation which is now associated with a strong decrease of circularity, in agreement with a deformation driven by a change of resting area (**Figure 2**). Moreover, we do not observe anymore the late rounding of cell, which was associated with the formation of the actomyosin ring in the WT. This demonstrates that cell extrusion in the pupal notum is poorly sensitive to global changes of the activation of MyoII. This result has to be confronted with our new results showing that Taxol treatment (which stabilises MTs) strongly impairs extrusion and slows down by a factor of 3 the speed of extrusion (**new Figure 6 I-K**). Altogether this demonstrates that the activation of MyoII is not an important rate limiting step of extrusion in the notum. The speed of extrusion strongly relies instead on the disassembly of MTs.

The new results on ROCK depletion is now described line 153:

“Importantly, while depleting ROCK (the main activator of MyoII[9]) by RNAi had a stringent effect on the epithelial morphology and blocked cytokinesis, it had no significant impact on the speed of extrusion (Figure S2B,C). In this context, the deformations occurring during extrusion were now associated with a strong decrease of circularity and the absence of cell rounding in the late phase (Figure S2C), confirming the link between the late rounding observed in WT extrusion and the increase of actomyosin (Figure 2J). Thus, global modulation of MyoII

activation has little impact on cell extrusion in the notum, in agreement with an extrusion regime poorly relying on changes in line tension.”

The results on Taxol are described line 310:

*“To confirm that MTs depletion is indeed an important rate-limiting step of extrusion, we stabilised MTs using Taxol injection at high concentration. While it did not totally block cell extrusion, it led to a global and drastic slowed-down of the speed of extrusion with variable durations (> 3 folds increase of extrusion duration, **Figure 6I-K, movie S21**). Moreover, the regime of deformation was now associated with a constant and significant increase of circularity (**Figure 6J**), similar to the deformations observed upon increase of line tension (**Figure 2G**), or upon *optoDronc* activation combined with *p35* expression (**Figure 4H**).”*

2) Figure 2:

Lines 127 onwards discussing the modelling: The purse-string cable that could form to help extrusion would be a feature of the surrounding cells, so would modelling an increase in line tension for the central cell really capture that?

Answer: We thank the reviewer for giving us a chance to clarify this point. The vertex model cannot distinguish easily the contribution of one cell or its neighbours regarding junctional tension as it models vertices mechanical equilibrium and includes a single line tension parameter for each junction (irrespective of the localisation of MyoII on one side or the other) combined with a perimeter line tension for each cell. To our knowledge, the relative contribution of the cortex of each neighbouring cell to junction tension is also very hard to sort experimentally. Anyway, since the purse string described in MDCK cells, pupal abdomen or pupal notum contains MyoII both from the dying cell and its neighbours, we do not believe it is essential to make this distinction. We now clarify this point in the main text to remove any ambiguity line 137:

“Note that the line tension parameter in the vertex model do not distinguish the contribution of the dying cell and its neighbours.”

3) Figure 3 C: from the still images I can see that the central extruding cell is showing a lowering of JupiterGFP levels, but equally, there are lots of cells just above that also show low MT levels, and other cells below that also show a lowering, but these are not apoptotic. Similarly, in Movie S8, the cell just above the extruding one also seems to lower MTs apically but that does not affect its apical area... How does this fit with the later experiments where the authors aim to show sufficiency of microtubule levels for increase or decrease of apical area. What effect on apical area do fluctuations of microtubule levels in the wild-type have? Either these changes in MT levels have an effect all the time, even in wt conditions where levels appear to fluctuate, or not.

Answer: It is indeed true that we observed systematic fluctuations of MT-reporter levels in the apical domain of cells (either with EB1-GFP or Jupiter-GFP). This is a combination of real biological fluctuations, but also sometimes artificially enhanced by the local projection procedure (which track the best focal plane locally and may fluctuate a bit from time point to time point, this is for instance visible in new **movie S13** with the fluctuating shadowing in the tubulin channel). First of all the amplitude of these fluctuations are not the same as the depletion we observed in extruding cells: on the order of 4-5% in control cells vs 15-30% in extruding cells (see **Figure R1**, at the end of this rebuttal and new **Figure 3D** subpanel and new **Figure S3-1 D'** subpanel). Moreover, these fluctuations are very transient and not

persistent relative to the constant decrease observed during extrusion (**Figure R1**, single curves). Accordingly, the average profile of EB1 or Jupiter GFP intensity is completely flat in non-extruding cells. What is then the cause of these fluctuations? Using cross-correlation we could show that EB1 and Jupiter GFP intensity are anti-correlated with cell perimeter (cell constriction is associated with MTs intensity increase) (**Figure R1**). This is coherent with our laser ablation experiment where we found that the reduction of cell apical area is associated with an increase of MT density (**Figure S4A-C**). This suggests that MTs tend to accumulate upon cell constriction/tension relaxation. This is in good agreement with two very recent preprints showing that cell compression leads to a stabilisation of MTs and a relative increase of their concentration[12, 13]. More importantly, these results and these two recent publications strengthen the fact that apical constriction/compression are not sufficient to disassemble MT, and that an active process of MT disassembly is required. This is exactly what happens during extrusion thanks to effector caspases. Of note, this is also in very good agreement with our previous published results showing that cell compression during mechanical competition will lead to cell extrusion only if caspase gets activated[6, 7]. We now mention this point about fluctuations at line 185:

*“While we observed some fluctuations of MTs intensity in the non-extruding cells, the amplitude of these variations were much more mild and non-persistent compared to the depletion observed during extrusion (see **Figure 3D inset, Figure S3-1 B’**).“*

For movies S8,9,10 it would be ideal to see the MT-related channel also by itself, please!

Answer: Thanks for this suggestion. We have provided all the individual channels in addition to the overlay which allows a better visualisation.

Another issue: the authors use overexpression of UAS-Shot-L(C)-GFP to visualise Shot (shown in Figure S3E). This isoform C lacks the actin-binding calponin homology domain (which the authors do not comment on at all here) and has been reported to bundle microtubules (Booth et al., DevCell 2014), so it does not report on wild-type Shot localisation and in fact changes microtubule behaviour when overexpressed. The best life Shot label is a GFP-exon trap that various labs hold (originally from Kyoto stock centre w[1118]; PBac[1, 2, 7, 8, 14]shot[KM0008] (DGRC number 109707), this labels one of the longest isoforms with all important domains. Or if a UAS construct is required, then the L(A) isoform also available from Bloomington contains both the actin-binding and microtubule-binding domains and at least in embryos and the germline looks most like anti Shot antibody labelling.

So the statement in lines 168-170 is not correct as UAS-Shot-L(C) will not report on endogenous Shot localisation.

Answer: We thank very much the reviewer for this very relevant point. Indeed we used the wrong Shot construct by mistake and therefore we could not conclude much about the dynamics of Shot during extrusion. We therefore used now the suggested reporter line from DGRC (number 109707). We actually saw very faint intensity in the pupal midline. We observed very light junctional pool in these cells that didn't seem to vary or relocalise over the course of cell extrusion (see **new Figure S3-1 F**). This observation confirms that it is unlikely that the observed MT-depletion is triggered by a decrease of Shot and/or the modulation of known MT organiser in the apical plane of the cell, although this is obviously limited by the poor quality of the signal.

A general question with regards to the changes in microtubules reported: Have the authors analysed microtubules in the whole volume of the cell? I .e. how certain are they it is really a reduction rather than a reorganisation as reported in other apically constricting cells (Booth et al., DevCell 2014)? The authors show that centrosomes remain apical, but does the association of microtubules with

centrosomes change? Are microtubules nucleated at centrosomes in these cells? If so, why are not more microtubules constantly being nucleated, what changes? Do centrosomes change?

Answer: This is indeed a very relevant alternative model which needs proper discussion. We looked previously at microtubules over the total cell volume using α -tubulin-mCherry. We tend to see a depletion throughout the cell. It is true however that this quantification remains difficult as it is very hard to distinguish the signal coming from the extruding cells and its neighbours, especially on the basal side where the geometry of cells becomes more irregular. To better support this point, we have characterised MTs dynamics using isolated cells expressing EB1-GFP and extruding (something relatively hard to catch actually since isolated clonal cells are rare). Using lateral view we could also see a diminution of signal both on the apical and basal side, while we could not see any obvious change of organisation nor a relocalisation of MTs (new **Figure S3-2 C,D**). Importantly, we now characterised MTs localisation throughout cells activating caspases with optoDronc (which allow the simultaneous activation in groups of cells, hence facilitating visualisation of MTs remodelling). In this situation we could unambiguously see a global depletion of MTs both on the apical and basal side cells, with no obvious relocalisation / reorganisation (new **Figure S3-2 E,E'**). Altogether, this clearly rules out a model based on MTs reorganisation/relocalisation during extrusion and really speak for a global disassembly.

These data are mentioned at line 200:

*“Accordingly, the disappearance of MT filaments is not restricted to the most apical domain and seems to occur throughout the cell, as visualised with α tub-mCherry (**Figure S3-2 A-B'**), or in single cells expressing EB1-GFP (**Figure S3-2 C-D**).”*

And also line 236:

*“Importantly, we confirmed in this condition that MTs depletion was occurring throughout the cell both on the apical and the basal side (**Figure S3-2, E,E'**).”*

Regarding the association of MTs with centrosomes, while we cannot have single MT resolution in our system, we can evaluate the orientation of MTs based on the direction of EB1 comets. Using high temporal resolution movies, we tracked a number of EB1 comets and showed that there is no obvious radial organisation and that MTs grows along the apical cortex with no obvious directionality (new **Figure S3-1 A,A'**). This is consistent with a medio-apical pool of MTs which are non-centrosomal, similar to the apical pool of MTs described in the pupal wing [15]. Importantly, this directionality does not seem to change during extrusion, we rather observed a progressive decrease of the number of comets (see new **movie S8** fusing fast acquisition movies at different phases of extrusion).

Line 176:

*“We tracked the orientation of MT growth in the medio-apical plane using the plus-end binding protein EB1, and found no obvious radial orientation of MTs (**Figure S3A**, check also the non-extruding cells in **movie S8**), in agreement with a non-centrosomal pool of MTs[15].”*

To conclude, we confirmed that the apical depletion correlates with a global depletion of MTs throughout the cells, and that there is no obvious relocalisation and/or reorganisation of MTs. The apical array of MTs are likely to be non-centrosomal and we do not find any obvious change of association of MTs with the centrosomes during extrusion. As pointed by Referee 1, we also now compared the timing of depletion of EB1 (as a proxy for polymerisation rate) and total pool of MTs (using sirTub) and could not find obvious delay even when using higher temporal resolution (30sec, the best we can do to catch several z-plane and catch extrusion

events, **new Figure S3-1 I-J**). This is all consistent with a model where effector caspases directly (or indirectly) deplete the pool of monomers which will affect both polymerisation and depolymerisation rates and lead to a progressive decrease of MTs throughout the dying cells. This point is discussed line 204:

*“We then checked whether this depletion was dominated by a change of polymerisation rate or depolymerisation rate of MTs by assessing putative lag-time between the change of EB1 comets (new growing MTs) and changes in the total pool of MTs (using sirTubulin). We could not detect any significant lag-time between the reduction of EB1 and total pool of MTs during extrusion (**Figure S3-1 I-J**), which could be compatible with a process affecting both the polymerisation and the depolymerisation rate (see below and **Discussion**).”*

The authors show a-tubulin-mCherry in Figure S3 H to illustrate a reduction of MTs from apical to basal in three optical sections. This label though does not seem to allow visualisation of microtubule filaments or bundles, it is just a hazy label. And whereas I can appreciate a reduction in intensity in the apical section, the lateral and basal section do not show clearly what is quantified in I, I'. In fact, even within the apical region there seem to be several other cells whose apical level of a-tubulin-mCherry is decreasing over the time-points shown, even though these cells are not dying or extruding. So how do we know that there are not just general fluctuation of apical MT intensity occurring? A better microtubule label should be used for the whole cell analysis, like SiR-tubulin or Jupiter-GFP.

Answer: As mentioned above, we have now better characterised the depletion of MTs throughout the extruding cells using EB1 and sirTub upon optoDronc activation. Moreover, we have clearly distinguished the fluctuations in non-extruding cells (in term of amplitude and duration) from the persistent and more significant decreases observed during extrusion (**Figure R1**). We are therefore very confident that MTs are globally reduced throughout the cell upon caspase activation.

4) Figure S4, A-B:

The authors test here whether the constriction of the apical area could be the inducer of the observed microtubule disassembly. From the images shown, I cannot tell which cell they are referring to and therefore I cannot tell if the images support their statement. The E-Cadherin label, especially for the highlighted section, is very hazy and does not clearly label cell outlines, it is also not shown in magnification. Furthermore, it is impossible for the reader to tell if the laser cutting led to any tissue relaxation, change in apical area, as no comparison of apical areas pre-and post-cut (in a way where individual cells can be assessed) is shown. This should be rectified to make the statement that apical constriction as a driver of MT disassembly can be excluded.

Answer: As suggested, we have now provided an image of the segmented cell before and after the cut as well as a crop with E-cad signal (**new Figure S4 A,C**). We already provided the quantification of the cell apical area based on segmentation of several movies (much more reliable than a serie of images from a single example) which clearly showed the rapid constriction of cells (~20% of cell apical area reduction, **Figure S4B**). This is also in good agreement with the negative correlation we observed between area fluctuations and EB1/Jupiter-GFP levels that we observed in non-extruding cells (**Figure R1**).

Figure 4:

Panels in A: the dotted line outlining the clone does not seem to match where the magenta label of the optoDronc can be seen. Again, various other cells also seem to show fluctuations of the sirTubulin label, so how do we know it is specific?

Answer: We apologise if the labelling was confusing. We highlighted only the cells that extruded in the clone during the time period shown in the panel. We have changed this to show the contour of full clone and pointed at the extruding cells using white arrows (**see new Figure 4B**). As for the fluctuations, we have discussed this point above. We also now provide a view of sir-tubulin on several z-plane of an optoDronc clones which unambiguously shows MT depletion throughout the cells (**new Figure S3-2 E,E'**)

Panels in D: Again, also other cells show random much higher intensity of the sirTubulin, and of the two outlined cells only one shows the increase, so how specific is this?

Answer: It is indeed correct that the enrichment of MTs is visible in only one of two cells. This is representative of the variability we observed, but what is clear and persistent is the absence of MT depletion in this context (to be compared with optoDronc alone, **new Figure 4A-D and Figure S3-2 E,E'**). Still, there is a tendency for a progressive increase of MT intensity (**Figure 4E,F**), in good agreement with what is observed upon laser cut and cell compaction, or based on the negative correlation between cell apical area and MTs intensity in non-extruding cells (**Figure R1**).

Panels in G: Again, there are many not small cells inside and outside of the clone that show equally high intensity of acetylated α -tubulin, so how can the reader assess that this is not random fluctuations of apical MT intensity levels?

Answer: Once again, there is indeed cell to cell variability. However we do observe very small cells in clones inhibited for caspase activation with relatively high levels of acetylated-tubulin, a combination that we almost never see in the WT. To better document this, we provide now a scatter plot showing cell apical area and tubulin intensity in WT and hid RNAi cells, as well as the histogram of distribution of cell apical area (**new Figure 4I,I'**). This clearly shows that there is more variability of cell size in hid RNAi context, with a fraction of very small cells not-present in the WT. This corresponds to cells with high acetylated tubulin levels in the cloud plot (bottom right). Our interpretation is that cells with such small apical area would normally undergo caspase activation and cell extrusion in the WT contexts. We have also modified the text to make this point more clear, line 243:

*“This is in agreement with the appearance of cells with low apical area and strong tubulin accumulation that we observed in hid-RNAi/caspase-inhibited clones (**Figure 4I,I'**), a combination that we do not observe in WT cells.”*

5) Figure 5:

Judging from panel A” (before injection) the apical microtubules usually span a lot of the apical area. The EB1-GFP labelled microtubules that have regrown after 125 sec seem to still be very short as shown in panel D. How do the authors think these microtubules are going to affect the apical cell area? What mechanism are they supposing would be at work to increase the area? I can see the argument that removing MTs might be a requirement for apical constriction to occur, as the microtubules might otherwise sterically block the process, but I cannot think of a mechanism that would allow microtubules to easily increase a cell’s apical area, especially not just newly nucleated ones that are still very short.

Answer: It is indeed correct that we do not seem to recover fully the control configuration of EB1 distribution in the colcemid + UV context. We would like to insist though that we cannot do a comprehensive characterisation of the MTs mesh in this context since most of our fluorophores gets rapidly bleached during the UV FLIP procedure. Still, we can clearly see comets that span the entire apical cell area which would be compatible with an effect on cell

apical size (see for instance top left cell in the **movie S15** bottom panel). This has to be put back in perspective with the relatively mild effect on apical area (5 to 10% increase), which would fit with a partial recovery of MTs. Last but not least, we believe we were quite careful in our interpretation and discussion to provide different mechanisms explaining how MT could affect apical area. At this stage, we cannot claim that they directly exert forces to stabilise apical area, so we would remain cautious about any interpretation connecting MTs morphology with cell apical area.

With regards to the LARIAT experiment: as the authors use overexpression of tagged α -tubulin, do they have any way of assessing what happens to endogenous tubulins present in these cells that most likely are all not affected? To be able to correctly interpret this experiment the authors need to know and show what happens to endogenous tubulin and microtubules overall. If they inject sirTubulin into these pupae, are apical microtubules actually disappearing in the LARIAT expressing and activated cells? This is essential to show! Also, the change in perimeter observed, though at one timepoint significant statistically, seems very small indeed. How can the authors exclude that it is not the effect of massive aggregates within the cells and some response to that that leads to the very mild change observed?

Answer: We are grateful to the reviewer for these important suggestions about new controls. First, we have now used clones only expressing a cytoplasmic UAS-GFP with the LARIAT system. This also triggers GFP aggregates, however there was no significant constriction of the cells (**new Figure 5H**), confirming that GFP aggregation per se cannot explain the transient constriction. To assess the impact on endogenous tubulin, we used the LARIAT system with sirTubulin as suggested by the referee. As expected, we only observe a mild depletion of tubulin (in agreement with the fact that only the tubulin-GFP will be depleted, **new Figure S5D**). This fits with the relatively mild constriction of cells that we observed. Admittedly, these are all very mild effects. This is however the best we could get so far. We actually tried to use in the notum a recently developed light sensitive analogue of Nocodazole (photostatin,[16]) to perform a local and time controlled depletion of MTs, but despite many tries we never managed to obtain a significant impact on MTs. This is in line with the much less potent effect of this drug (10x less) compared to Colcemid, and the fact that we already have to use high concentrations of Colcemid (2.5mM) for injection to see an effect in the pupae. Since the mild effect of LARIAT is reproducible, we believe it is worth inserting this data. The new results are in **Figure 5H** and **S5D** and described in the result part as followed line 277:

*“Accordingly, blue light exposure led to rapid clustering of GFP α -Tubulin, a mild reduction of sir-Tubulin signal (marking the total tubulin pool, **Figure S5D**) and accordingly a mild but reproducible reduction of cell apical area without clear modulation of MyoII (**Figure 5G-H, movie S16**). This apical area reduction was specific of tubulin sequestration since it was not observed upon clustering of a cytoplasmic GFP by LARIAT (**Figure 5H**).”*

Figure 6: The authors state that spastin-activation in clones appears to increase extrusions. There seems to be a lot of variability, though, from pupa to pupa, in terms of the number of extrusions. The overall numbers in between control and experiment in A seems quite different, and even more so, there seems to be a huge difference in the number of extrusions between the controls in A and in C. Analysed were 3 pupae with spastin clones and 2 pupae as control. With such differences in numbers, at least as it appears from the images shown, more pupae need to be analysed. Also, with regards to spastin-expression: in our and other labs' hands the microtubule depletion induced by spastin overexpression in different tissues tends to not be complete, in the sense that not all cells that express spastin will show a decrease in microtubules (and it is unclear why that is, but it has been observed by several labs). So have the authors analysed how effective the clonal expression

of spastin is on microtubules? Have they stained for α -tubulin and acetylated α -tubulin upon spastin induction? This should be shown.

Answer: Thank you for suggesting this important control and for giving us a chance to clarify some of the results. First, quantification from **Figure 6B** and **6D** were not made the same way given the different nature of the perturbation. In **B,B'** we differentiate the number of extrusion in the midline region (**B**, a region with high rates of extrusion) and outside midline (**B'**, a region with low rate of extrusion). The numbers we obtained are actually consistent with what was previously published [6, 7] (~20-30% in the midline, 1-2% outside midline previously usually calculated over 600 min, here the measurements were made over 1000 min). In **D**, the quantification was made in the whole notum irrespective of the region since cell extrusion is normally homogeneously inhibited by hid RNAi (see [7]). Also the total duration of the movie analysed are different between these two experiments (1000 min for Spastin, 500 min for colcemid, we have to restrict the colcemid analysis time window to the time where MT are clearly depleted and before the tissue starts to be severely affected). It is therefore not really relevant to compare these two values. We have clarified this in the figure legend to avoid any confusion.

Regarding Spastin, it is indeed a relevant point. This should not change our conclusion though since we already see a very clear increase of the rate of extrusion and a lot of ectopic cell extrusion in regions where we normally see none (even if MT depletion is variable and partial). We have now checked the depletion of MTs in UAS-Spastin clones using *siTubulin*, and confirmed that there was a very clear depletion of MTs. We did observe some heterogeneity within clones which actually matches the variability of the levels of expression of UAS-RFP. This could be explained by the inherent variability of the Gal4 combined with Gal80ts. We have included this new data in the manuscript (new **Figure S6A**)

If colcemid injection, even in the wt, increases extrusions across the notum, as the authors seem to show in this figure, then interpreting what they show in Figure 5 with colcemid injections is much harder as there must be so much unusual extrusions going on. So it is hard to know how individual cells will respond in a tissue with such changes. This effect of colcemid should have been mentioned in the discussion of Figure 5.

Answer: Thank you for raising this point. We realise this may have been confusing since the two experiments are not at all performed on similar timescales. In the experiments of **Figure 6** we injected colcemid, waited 1h30 to observe clear MT depletion, and image the tissue over several hours (500 min). In **Figure 5**, the same procedure is applied, but the UV inactivation and cell shape changes are taking place on the timescale of few minutes (much less than the characteristic time of extrusion, from 20 to 30min). Thus, we can safely neglect the impact of extrusion in this experiment. To clarify this we clearly indicates the time scale in the scheme of **Figure 5A,A'**.

A more general question: by what mechanism do the authors assume does a depletion of microtubules lead to cell extrusion? So far the authors were arguing that microtubule depletion is permissive for apical area reduction and extrusion in the wild-type, but with Figure 6 they seem to argue that it is sufficient to drive it. How? What happens once microtubules are gone? Because this is clearly not what happens in many other tissues, including the embryonic epidermis at various stages, when spastin is overexpressed here. So what is different about notum cells, what is the (presumed, hypothesised) molecular mechanism?

Answer: This is indeed a very relevant question that deserves more space in the discussion. For the pupal notum, our results are compatible with the impact of MT (direct or indirect) on the resting area of cells. As mentioned in the discussion, different mechanisms could be responsible for this effect, either through a direct impact of MTs on cell mechanics, or an indirect effect on actomyosin contractility, or on the viscosity of the cytoplasm or even on nucleus properties and positioning. In normal conditions, the negative correlation between MTs and cell constriction (**Figure R1**) could constitute a simple stabilising mechanism that prevents too much cell deformation. This is in good agreement with the very recent publications showing that compression can stabilise MTs[12, 13], and the local increase of MTs concentration we observed upon laser severing which is then followed by a re-expansion of cell apical area (**Figure S4**). Upon global depletion of MTs, this stabilising mechanism would be gone and large area fluctuations would lead to spontaneous cell extrusion, even in absence of caspase activation. Accordingly, previous 3D vertex modelling have shown that certain topology or reduction of apical area below a threshold value can lead to mechanical instability and spontaneous extrusion[17].

We now mention this point in the discussion line 411:

“The stabilisation of MT upon cell constriction may help to buffer variations of the cell apical area despite the fluctuations of line tension[18] or external mechanical constrains. In absence of MTs, this negative feedback would be gone, hence allowing large fluctuations of cell area and the appearance of spontaneous extrusion driven by local mechanical instabilities[17]. Interestingly, depletion of MTs in other systems (e.g.: the fly embryo[19]) does not seem to have the same impact on epithelial stability and cell extrusion, suggesting that the impact of MTs on epithelial cell stabilisation may be context-dependent.”

Regarding the difference with the fly embryo, several explanations could be put forward: first, the extend of MTs depletion upon Spastin expression might be different (along this line, there is still a very significant pool of acetylated tubulin in the salivary gland upon Spastin expression, Figure 7 of [19], while we hardly see any MTs remaining upon Spastin induction in the notum), secondly, the time scales might also be very different (1 or 2 hours in the embryo, depletions over >10 hours in the notum), finally the core mechanical properties (ECM, cell shape, relative tension of the apical lateral and basal surface, nuclear properties) might completely change the relative contribution of MTs to epithelial stability (similar to the impact of ROCK inhibition on different morphogenesis processes). We have now included a more thorough discussion on the mechanism of cell apical area stabilisation by MTs in the discussion and the relevance in other tissue by mentioning that the contribution of MTs might be different in the fly embryo (see quoted text above).

With regards to figure panels in this figure:

The Cadherin cell outline label shown in E and E' is really not very good, why is this? It is impossible for the reader to see the cell outlines, so impossible to judge how accurately the dotted line represents them. Also, what is happening to the cells surrounding the extruding cell in panels in G? They seem to massively increase their apical area judged by the much better E-Cad label shown here. What is going on?

Answer: It is true that E-cad signal is not really good in this context. This comes from the condition (colcemid injection + hid-RNAi) combined with the properties of E-cad-tdTomato which tends to accumulate in endosomes (contrary to E-cad-GFP which is quenched by low pH). While the contours are indeed hard to see on the stills, these are more clear when watching the movies (where one can use the temporal profile to see better the contour). We do not have much better picture to propose so we will stick to what we have here, but now

point to cell centre and only show the contour on the GC3Ai channel to let the readers see more properly the signal. What matters in this context is anyway the quantification provided in **Figure 6F**. As for the enlargement of cells in **Figure 6G**, this is explained by the entry in mitosis which fails to progress because of colcemid injection. These movies were taken at a stage where there is a large number of cells entering in mitosis. We now explain this in the legend. The quality of the movie is better also because this is taken on much shorter timescale (thanks to the time-controlled activation of optoDronc) and at a different magnification (40X in **E,E'**, 100X in **G**).

7) Discussion and title:

The authors state at the beginning of the discussion: 'This is, to our knowledge, one of the first descriptions of a role of MTs in the initiation of cell extrusion independently of MyoII.' This is a somewhat confusing way of phrasing the results. MTs do not have 'a role...in the initiation..', rather, what the authors aim to show is that they need to be gone for extrusion to happen. This phrasing somehow implies an active role of MTs, whereas what the authors argue is that their disassembly is permissive.

Answer: This is indeed a potentially confusing formulation. We indeed aim to show that MT depletion is permissive for extrusion. As suggested we have replaced the mentioned sentence by the following, line 324:

"This is, to our knowledge, one of the first descriptions of a permissive role of MTs depletion in the initiation of cell extrusion independently of MyoII."

With regards to the title 'Microtubule disassembly by caspases is the rate-limiting step of cell extrusion', I think it would be more correct to say 'Microtubule disassembly by caspases is a rate-limiting step of cell extrusion'. The authors have not analysed or shown here whether the microtubule disassembly is the only rate-limiting step.

Answer: This is totally correct. Indeed we cannot exclude that alternative factors modulate the rate of cell extrusion. Yet, since our recent results on Taxol shows that MTs stabilisation has a very significant impact on extrusion speed (much more than ROCK depletion), and based on our results of the permissive effect of colcemid on extrusion (despite caspase inhibition), we think it is fair to state the MTs depletion is an important rate limiting step of extrusion. As such we have changed the title for the following:

'Microtubule disassembly by caspases is an important rate-limiting step of cell extrusion'.

Figure R1

Figure R1: Variations of Microtubule levels in non-extruding cells in the notum.

A: Correlation between perimeter variation and EB1-GFP intensity in non-extruding cells. The correlation coefficient (r^2) is 0.371. $N = 1220$ data points. **B:** Average cell perimeter (black) and EB1-GFP (green) in control non-extruding cells. $N = 27$ cells. Light colour shows s.e.m.. **B':** Example of a single representative cell showing variations of its perimeter (black) and EB1-GFP intensity (green). **C:** Average and normalised cross correlation of cell perimeter against EB1-GFP intensity. Maximum of correlation is obtained at $t = -2\text{min}$ (n.s from $t = 0\text{min}$). Note that this correlation is negative (while it is positive for extruding cells). Light colour shows s.e.m.. $N = 27$ cells. **D:** Average cell perimeter (black) and Jupiter-GFP (green) in control non-extruding cells. $N = 21$ cells. Light colour shows s.e.m.. **D':** Example of a single representative cell showing variation of its perimeter (black) and Jupiter-GFP intensity (green). **E:** Average and normalised cross correlation of cell perimeter against Jupiter-GFP intensity. Maximum of correlation is obtained at $t = 0\text{min}$. Note that this correlation is negative. Light colour shows s.e.m.. $N = 21$ cells.

Rebuttal references

1. Rosenblatt, J., Raff, M.C., and Cramer, L.P. (2001). An epithelial cell destined for apoptosis signals its neighbors to extrude it by an actin- and myosin-dependent mechanism. *Curr Biol* *11*, 1847-1857.

2. Kuipers, D., Mehonic, A., Kajita, M., Peter, L., Fujita, Y., Duke, T., Charras, G., and Gale, J.E. (2014). Epithelial repair is a two-stage process driven first by dying cells and then by their neighbours. *J Cell Sci* *127*, 1229-1241.
3. Michael, M., Meiring, J.C.M., Acharya, B.R., Matthews, D.R., Verma, S., Han, S.P., Hill, M.M., Parton, R.G., Gomez, G.A., and Yap, A.S. (2016). Coronin 1B Reorganizes the Architecture of F-Actin Networks for Contractility at Steady-State and Apoptotic Adherens Junctions. *Dev Cell* *37*, 58-71.
4. Teng, X., Qin, L., Le Borgne, R., and Toyama, Y. (2017). Remodeling of adhesion and modulation of mechanical tensile forces during apoptosis in *Drosophila* epithelium. *Development* *144*, 95-105.
5. Simoes, S., Oh, Y., Wang, M.F.Z., Fernandez-Gonzalez, R., and Tepass, U. (2017). Myosin II promotes the anisotropic loss of the apical domain during *Drosophila* neuroblast ingression. *J Cell Biol* *216*, 1387-1404.
6. Levayer, R., Dupont, C., and Moreno, E. (2016). Tissue Crowding Induces Caspase-Dependent Competition for Space. *Curr Biol* *26*, 670-677.
7. Moreno, E., Valon, L., Levillayer, F., and Levayer, R. (2019). Competition for Space Induces Cell Elimination through Compaction-Driven ERK Downregulation. *Curr Biol* *29*, 23-34 e28.
8. Valon, L., Davidović, A., Levillayer, F., Villars, A., Chouly, M., Cerqueira-Campos, F., and Levayer, R. (2021). Robustness of epithelial sealing is an emerging property of local ERK feedback driven by cell elimination. *Developmental Cell* *56*, 1-12.
9. Levayer, R., and Lecuit, T. (2012). Biomechanical regulation of contractility: spatial control and dynamics. *Trends Cell Biol* *22*, 61-81.
10. Schott, S., Ambrosini, A., Barbaste, A., Benassayag, C., Gracia, M., Proag, A., Rayer, M., Monier, B., and Suzanne, M. (2017). A fluorescent toolkit for spatiotemporal tracking of apoptotic cells in living *Drosophila* tissues. *Development* *144*, 3840-3846.
11. Roll-Mecak, A., and Vale, R.D. (2005). The *Drosophila* homologue of the hereditary spastic paraplegia protein, spastin, severs and disassembles microtubules. *Curr Biol* *15*, 650-655.
12. Li, Y., Kučera, O., Cuvelier, D., Rutkowski, D.M., Deygas, M., Rai, D., Pavlovič, T., Vicente, F.N., Piel, M., Giannone, G., et al. (2022). Compressive forces stabilise microtubules in living cells. *bioRxiv*, 2022.2002.2007.479347.
13. Ju, R.J., Falconer, A.D., Tang, C.K.X., Dean, K.M., Fiolka, R.P., Sester, D.P., Nobis, M., Timpson, P., Lomakin, A.J., White, M.D., et al. (2022). A Microtubule Mechanostat Enables Cells to Navigate Confined Environments. *bioRxiv*, 2022.2002.2008.479516.
14. Acharya, B.R., Nestor-Bergmann, A., Liang, X., Gupta, S., Duszyc, K., Gauquelin, E., Gomez, G.A., Budnar, S., Marcq, P., Jensen, O.E., et al. (2018). A Mechanosensitive RhoA Pathway that Protects Epithelia against Acute Tensile Stress. *Dev Cell* *47*, 439-452 e436.
15. Singh, A., Saha, T., Begemann, I., Ricker, A., Nusse, H., Thorn-Seshold, O., Klingauf, J., Galic, M., and Matis, M. (2018). Polarized microtubule dynamics directs cell mechanics and coordinates forces during epithelial morphogenesis. *Nat Cell Biol* *20*, 1126-1133.
16. Borowiak, M., Nahaboo, W., Reynders, M., Nekolla, K., Jalinot, P., Hasserodt, J., Rehberg, M., Delattre, M., Zahler, S., Vollmar, A., et al. (2015). Photoswitchable Inhibitors of Microtubule Dynamics Optically Control Mitosis and Cell Death. *Cell* *162*, 403-411.
17. Okuda, S., and Fujimoto, K. (2020). A Mechanical Instability in Planar Epithelial Monolayers Leads to Cell Extrusion. *Biophys J* *118*, 2549-2560.
18. Curran, S., Strandkvist, C., Bathmann, J., de Gennes, M., Kabla, A., Salbreux, G., and Baum, B. (2017). Myosin II Controls Junction Fluctuations to Guide Epithelial Tissue Ordering. *Dev Cell* *43*, 480-492 e486.
19. Booth, A.J.R., Blanchard, G.B., Adams, R.J., and Roper, K. (2014). A dynamic microtubule cytoskeleton directs medial actomyosin function during tube formation. *Dev Cell* *29*, 562-576.

REVIEWERS' COMMENTS

Reviewer #1 (Remarks to the Author):

The manuscript has been revised along the lines suggested by the different referees. Overall, it has considerably improved and the main claims are now sufficiently supported by experimental data. I now fully support publication of this manuscript.

Reviewer #2 (Remarks to the Author):

In this revision, the authors adequately responded to all of my concerns. This study will be a great addition to the field. Congratulation!

Reviewer #3 (Remarks to the Author):

Response to the authors answer to my original point 3):

In this revised version of the manuscript the authors have made an excellent effort in answering the concerns and questions raised by all three reviewers, which is very much appreciated.

The added information makes the paper clearer and better, and clearly a very interesting contribution to the fields of morphogenesis.

I have one niggle left that would be good to clarify even more, please.

With regards to natural changes in microtubule levels/fluctuations of amount of microtubules in all cells and how this impacts on apical cell area:

I was not suggesting that apical constriction would lead to microtubule disassembly. Rather, I was asking about fluctuation of microtubule levels in all cells, because the authors claim that disappearance/reduction of microtubules is sufficient to induce cell constriction. If this is true, shouldn't then apical area fluctuate with amount of apical microtubules, and the correlation should be that smaller apical area correlates positively with reduced amount of microtubules. What the authors discuss in cite in their response here is that in fact apical constriction correlates with an increase in microtubule labelling...

So I am still confused about the sufficiency argument. So is only a complete ectopic depolymerisation/loss of microtubules sufficient for extrusion? But from their Lariat experiment they now state that the reduction in tubulin is only mild and 'this fits with the relatively mild constriction of cells that we observed', as they say in their response. So this would suggest there should be an effect of even mild tubulin/microtubule fluctuation that should correlate in the way 'less microtubules correlates with smaller area'.

Answer to reviewers

“Microtubule disassembly by caspases is an important rate-limiting step of cell extrusion”

General comments

We would like to thank all the reviewers for their thoughtful comments and acknowledging the importance of this work. We provide here an answer to the last point raised by Reviewer 3.

Reviewer comments in “Calibri”

Answer in “Arial”

Reviewer #3 (Remarks to the Author):

Response to the authors answer to my original point 3): In this revised version of the manuscript the authors have made an excellent effort in answering the concerns and questions raised by all three reviewers, which is very much appreciated. The added information makes the paper clearer and better, and clearly a very interesting contribution to the fields of morphogenesis. I have one niggle left that would be good to clarify even more, please. With regards to natural changes in microtubule levels/fluctuations of amount of microtubules in all cells and how this impacts on apical cell area: I was not suggesting that apical constriction would lead to microtubule disassembly. Rather, I was asking about fluctuation of microtubule levels in all cells, because the authors claim that disappearance/reduction of microtubules is sufficient to induce cell constriction. If this is true, shouldn't then apical area fluctuate with amount of apical microtubules, and the correlation should be that smaller apical area correlates positively with reduced amount of microtubules. What the authors discuss in cite in their response here is that in fact apical constriction correlates with an increase in microtubule labelling... So I am still confused about the sufficiency argument. So is only a complete ectopic depolymerisation/loss of microtubules sufficient for extrusion? But from their Lariat experiment they now state that the reduction in tubulin is only mild and 'this fits with the relatively mild constriction of cells that we observed', as they say in their response. So this would suggest there should be an effect of even mild tubulin/microtubule fluctuation that should correlate in the way 'less microtubules correlates with smaller area'.

Answer:

Thank you for acknowledging the improvement of the manuscript. This last question is indeed very important, and we would like to clarify this point here. It is indeed true that it may appear surprising to find a negative correlation between cell apical area and MTs intensity in the control cells, while this correlation is positive during extrusion or during our perturbative experiments. We believe this is related to the multiple feedbacks we were invoking in our previous explanations. They probably make the interpretation of the cross-correlation curves difficult in the non-extruding cells. Indeed, MTs stabilise apical area, but cell constriction can also increase MTs concentration. Therefore, their peak depends on which effect is dominant in each situation and with what time delay. Accordingly, the correlation peak we observed are relatively small in these conditions (~-0.2, versus 0.6-0.7 for the correlation between sirTub and cell apical perimeter during extrusion). This is also in agreement with the single cross-correlation curves which shows either positive or negative correlations at t_0 depending on the cell (see **Figure R1** below). The averaged slight negative correlation observed during spontaneous fluctuations of cell apical area simply suggests that these area fluctuations are

most likely slightly dominated by actomyosin fluctuations and/or the fluctuations of tension in the neighbouring cells. The situation becomes very different when there is an active remodelling of MTs, could it be during extrusion, drug treatment (colcemid +UV) or the LARIAT experiment. Here we impose a rapid and sustained change of MTs which will impact cell shape and reveal the stabilising role of MTs. Accordingly, the cross-correlation becomes clearly positive in these conditions with much less variability (see for instance **Figure R2** for the single curves of cross-correlation between sirTub and perimeter during optoDronc extrusion). All in all, based on the experiments with colcemid, LARIAT, spastin, we are confident to claim that MT depolymerisation is indeed sufficient to trigger cell extrusion, at least in the pupal notum. Since the interpretation of cross-correlation curves in the non-extruding cells are difficult to interpret and are not really useful for our claim, we would prefer not to include them in the manuscript. We hope the reviewer will agree with this point and that the explanations we provide here are helpful.

Figure R1: Single cross-correlation curves between cell apical perimeter and EB1 intensity in non-extruding cells. Note the large variations in the peak at 0 time lags (some are positive, other negatives).

Figure R2: Single cross-correlation curves between cell apical perimeter and sir-Tubulin intensity upon extrusion induced by optoDronc activation. Note the very reproducible positive peak at t0.